# Symmetric and asymmetric DNA N6-adenine methylation regulates different biological responses in Mucorales

Carlos Lax [1], Stephen J. Mondo [2,3,4], Macario Osorio-Concepción [1], Anna Muszewska [5], María Corrochano-Luque [6], Gabriel Gutiérrez [6], Robert Riley [2], Anna Lipzen [2], Jie Guo[2], Hope Hundley[2], Mojgan Amirebrahimi[2], Vivian Ng [2], Damaris Lorenzo-Gutiérrez[1], Ulrike Binder [7], Junhuan Yang[8], Yuanda Song[9], David Cánovas [6], Eusebio Navarro [1], Michael Freitag[10], Toni Gabaldón [11,12,13,14], Igor V. Grigoriev [2,4,15], Luis M. Corrochano [6] ✉, Francisco E. Nicolás [1] ✉ & Victoriano Garre [1] ✉

DNA N6-adenine methylation (6mA) has recently gained importance as an epigenetic modification in eukaryotes. Its function in lineages with high levels, such as early-diverging fungi (EDF), is of particular interest. Here, we investigated the biological significance and evolutionary implications of 6mA in EDF, which exhibit divergent evolutionary patterns in 6mA usage. The analysis of two Mucorales species displaying extreme 6mA usage reveals that species with high 6mA levels show symmetric methylation enriched in highly expressed genes. In contrast, species with low 6mA levels show mostly asymmetric 6mA. Interestingly, transcriptomic regulation throughout development and in response to environmental cues is associated with changes in the 6mA landscape. Furthermore, we identify an EDF-specific methyltransferase, likely originated from endosymbiotic bacteria, as responsible for asymmetric methylation, while an MTA-70 methylation complex performs symmetric methylation. The distinct phenotypes observed in the corresponding mutants reinforced the critical role of both types of 6mA in EDF.

Early-diverging fungi (EDF) are a diverse group comprising up to 17 different phyla[1–4], which are poorly studied compared to the Dikarya (Ascomycota and Basidiomycota)[5]. However, research on EDF has provided significant insights into the evolution, ecology, and adaptation of fungi[6]. They include species with increasing clinical and biotechnological relevance. Some are phytopathogenic[7,8], while others establish symbiosis with plants, like Arbuscular mycorrhizal fungi (Glomeromycota)[9,10]. Some Mucorales (mainly *Rhizopus*, *Mucor,* and *Lichtheimia*) are opportunistic human pathogens[11,12], whereas some chytrids cause chytridiomycosis, a fungal infection contributing to the global extinction of amphibian species globally[13,14]. Anaerobic species

(*Neocallimastigomycota*) found in the digestive tract of some animals are interesting as enzyme producers[15,16], while other are of interest to the food industry and biofuel research, given their high level of lipid production (*Mucor, Blakeslea,* and *Mortierella*)[17–20]. EDF display unique characteristics within the fungal kingdom, including flagella, an actin cytoskeleton, cobalamin utilization, and cholesterol-containing membranes, which can be traced back to the Opisthokont ancestor shared with animals[21–24].

Despite being phylogenetically closer to animals, EDF share characteristics with evolutionary distant eukaryotic lineages like ciliates and green algae, including flagellation and a particular landscape

of DNA modifications[25]. In addition to bacteria, N6-methyladenine (6mA) is found at high levels in EDF, ciliates and green algae[26-30]. In prokaryotes, 6mA participates in genome defense (restriction/modification system), DNA replication and repair, nucleoid segregation, and gene expression regulation[31-33]. Although the relevance of DNA 6mA in multicellular eukaryotes has been controversial due to relatively low levels, especially in animals, 6mA is abundant and shares features with respect to its distribution and location in the three above-mentioned groups of unicellular eukaryotic organisms[34].

Unfortunately, little is known about the regulatory implications of epigenetic modifications in EDF. The role of posttranscriptional regulation mediated by the RNAi machinery has been extensively characterized in the pathogenic fungus *Mucor lusitanicus*[35-38] (hereafter *Mucor*) including its implication in environmental adaptation through the generation of transient epimutants[39-42]. However, few studies have focused on understanding the biological roles of other epigenetic modifications in these fungi[30,43-46], contrasting with Dikarya, plants, and animals. EDF include representatives with variable levels of 5-methylcytosine (5mC) and 6mA DNA modifications[30,47,48] highlighting the potential for exploring the evolution of DNA modifications. In this study, we survey the methylomes of 62 species across most phyla of EDF and focus on the Mucorales *Mucor* and *Phycomyces blakesleeanus* (hereafter *Phycomyces*), each displaying different extremes with respect to 6mA and 5mC utilization. By coupling 6mA and 5mC distributions with gene expression analyses, we characterize the role of 6mA and its relationship with 5mC, both with respect to regulation of gene expression and in response to environmental and developmental cues. Additionally, we identify the adenine methyltransferase machineries responsible for both symmetric and asymmetric 6mA. Phenotypic characterization of 6mA methyltransferase mutants reveals a crucial regulatory role for both types of 6mA modifications in *Mucor*, and likely other EDF.

## Results

### Epigenomic 6mA landscapes vary across early-diverging fungi
6mA, a less frequently studied DNA modification, has been found in high and reliable levels in green algae, ciliates and EDF[25,30,45]. The original characterization of 6mA in 16 fungal species, with 10 EDF[30], has been extended to 62 different fungal species (56 EDF), (Supplementary Table 1, Fig. 1) by expanding the number of species in most phyla and including two Zoopagomycota and one Monoblepharomycota species. Although EDF were initially considered 6mA-rich organisms, our analysis has unveiled variability among the different fungal phyla. Species from the Mucoromycota, Mortierellomycota, Kickxellomycota, Zoopagomycota, Neocallimastigomycota, and Basidiobolomycota displayed the highest 6mA levels (0.8 to 3.8%) (Supplementary Table 1). Conversely, Chytridiomycota, Monoblepharomycota, Blastocladiomycota, and even Glomeromycota, exhibited reduced 6mA levels (<0.3%), similar to Dikarya. Interestingly, while Mucoromycota representatives have highly methylated genomes (1.6% of all A's are methylated genome-wide, on average), this phylum displayed marked variability, with species like *Mucor* and *Blakeslea trispora* exhibiting <1% of 6mA (Supplementary Table 1, Fig. 1). Similarly, chytrids, which generally showed reduced 6mA levels, had exceptions with species like *Triparticalcar arcticum*, *Powellomyces hirtus*, and *Fimicolochytrium jonesii*, which contain high (>1%) 6mA levels (Supplementary Table 1, Fig. 1). These results revealed substantial variation in 6mA content, not just between EDF and Dikarya but also within EDF phyla and even species.

### *Mucor* and *Phycomyces* are two early-diverging fungi with distinct methylomes
To study the potential epigenetic regulation in EDF and species differences, we selected *Mucor* and *Phycomyces* for further analysis. *Mucor* was chosen because it has the largest number of available

molecular tools within EDF[49], while *Phycomyces* is a model organism for light perception and development in EDF[50]. Additionally, these two species exhibited contrasting methylomes, with *Phycomyces* displaying high 6mA levels (Supplementary Table 1, Fig. 2a, c, d, and Supplementary Fig. 2), and *Mucor* showing a reduced proportion of 6mA (Supplementary Table 1, Fig. 2a, c, d, and Supplementary Fig. 1). Since EDF epigenomes are mostly unexplored, we used PacBio Single Molecule Real Time (SMRT) sequencing to generate the first genome assembly of *Phycomyces* L51 strain, a blind mutant, along with updated versions of *Phycomyces* UBC21 and *Mucor* CBS 277.49, all available at MycoCosm[51]. The new *Mucor* CBS 277.49 v3 assembly reduced the number of scaffolds from 26 to 12, and we successfully identified twelve telomeres (TTAGGG repeats), including three telomere-to-telomere scaffolds (Supplementary Fig. 1). Genome-wide 6mA profiling from PacBio SMRT reads reported 1.13% of 6mA in the *Phycomyces* genome (NRRL1555−mycelium grown in the dark) and 0.26% in the genome of *Mucor* (CBS 277.49−mycelium grown in the dark) (Supplementary Table 2, Fig. 2d). To validate these values, we performed HPLC-MS/MS analysis with DNA from these and related strains. The 6mA levels detected in this analysis for *Phycomyces* NRRL155 and UBC21 were 1.13% and 1.04%, respectively, while they ranged between 0.05 and 0.25% for *Mucor* strains, confirming PacBio-SMRT results (Fig. 2d). We also corroborated these differences through dot blot using an antibody specific for 6mA (Supplementary Fig. 3A). The differences in 6mA levels were noticeable even with an isoschizomer digestion assay, in which genomic DNA was digested with *Dpn*I (cleaves methylated G6mATC) and *Dpn*II (blocked by 6mA). A clear degradation pattern was visible in *Phycomyces* DNA digested with *Dpn*I, but not in *Mucor* (Fig. 2c), indicating that 6mA in the AT context was more frequent in the *Phycomyces* genome.

*Mucor* and *Phycomyces* differed not only in 6mA levels but also at the methylation ratio of each site. Almost every 6mA site in the *Phycomyces* genome had a methylation ratio of 1 (methylated reads / total reads), whereas this ratio was around 0.5 for *Mucor* sites (Fig. 2e), similar to *R. irregularis*[45]. 6mA was abundant in *Phycomyces* but scarce in *Mucor*, yet broadly distributed across both genomes (Fig. 2a, Supplementary Fig. 1 and Supplementary Fig. 2). Although it followed a similar pattern in its distribution over genomic features, 6mA in *Phycomyces* was more frequently detected in gene-rich regions (specifically towards its 5' end) and was devoid in repeat-rich regions (Fig. 2a, b, Supplementary Fig. 2).

Only 0.32% of 6mA sites in *Mucor* were symmetric, whereas most in *Phycomyces* were symmetric (91.72%) (Supplementary Table 2 and Fig. 2f). Linked to this, the most enriched 6mA motif in *Phycomyces* was VATB (V: A, C or G and B: G, T or C) with 97.70% in ApT dinucleotides (Supplementary Table 2, Fig. 2g), as *Dpn*I digestion suggested. In *Mucor*, the preferentially methylated motif was AAA6mACA, with most of the methylated sites flanked by A/G (5') and C/G (3'). This motif, unlike the most frequent motif in EDF, was similar in other complex eukaryotic organisms, including Dikarya[30]. Methylated sites in *Mucor* also resembled the CA6mAC and AA6mAGA motifs in plants[52,53]. Also, there were several G6mAGG methylation sites in *Mucor* that coincided with the motif in nematodes[54], which is recognized by the METL-9-DVE-1 complex[55].

6mA was not the only DNA modification that differs between *Mucor* and *Phycomyces*. Mucoromycota overall exhibit reduced 5mC levels[30,47], but the level in *Phycomyces* was exceptionally high[47,48]. Analysis of weighted methylation levels by cytosine context (CG, CHG, and CHH) revealed similar levels for *Mucor* (0.3%) but enrichment in CG context for *Phycomyces* (3.4%, 0.5%, and 0.9%, respectively). Over 25% of 5mC sites located at CpG dinucleotides with 200x enrichment, considering their genomic frequency (Supplementary Table 2, Supplementary Table 3, Supplementary Fig. 3B, C). Unlike 6mA, 5mC sites in *Phycomyces* were almost exclusively in repeat-rich regions (Fig. 2b) similar to *Linderina pennispora* (Zoopagomycota)[30], now expanding

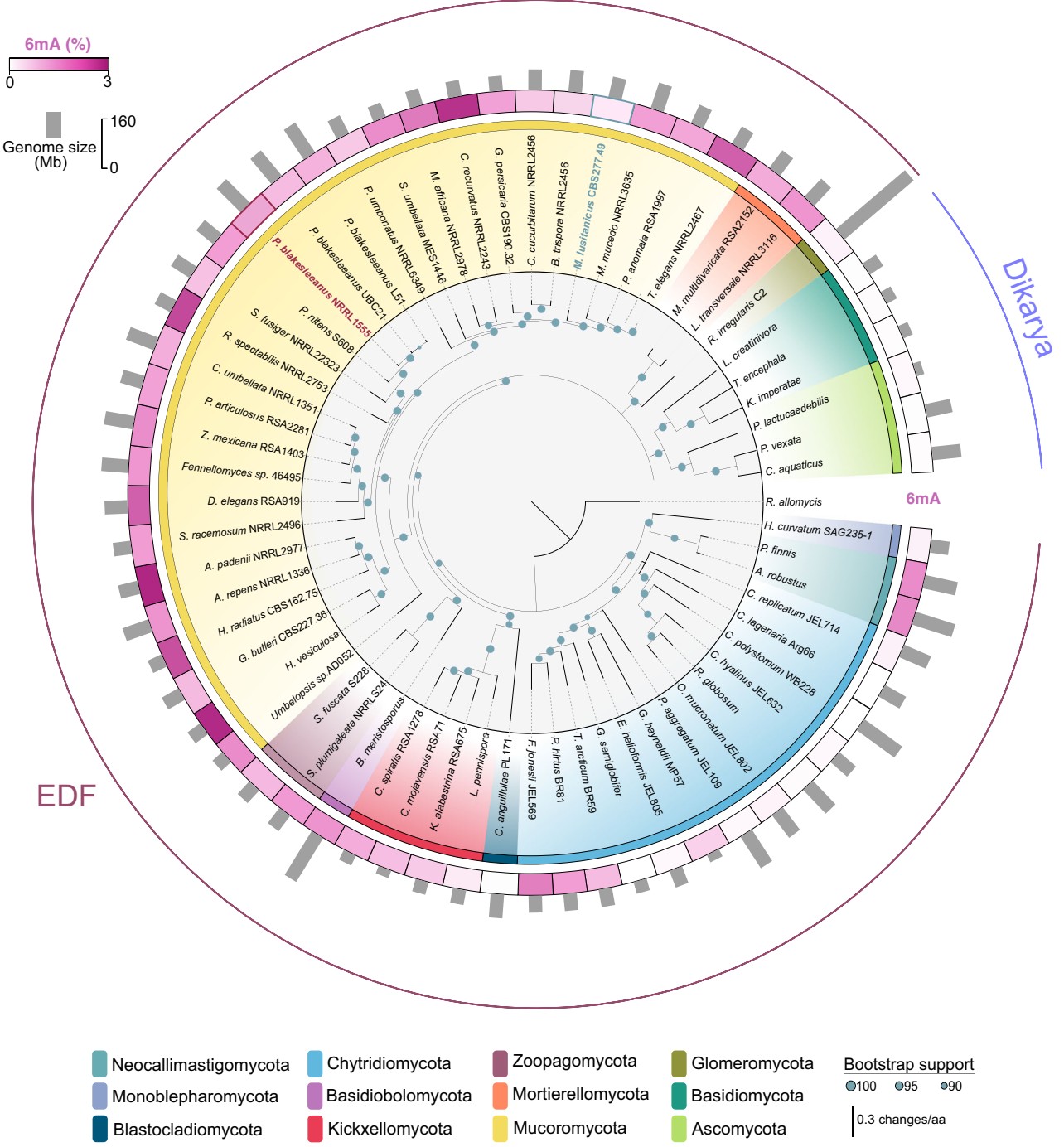

**Fig. 1 | 6mA level variation in the fungal kingdom.** The genomic levels of 6mA (% of methylated adenines) vary dramatically between the different fungal phyla. Also, marked differences are found between species belonging to the same phylum. The inner species tree shows the phylogenetic relationship between 62 fungal representatives for the different fungal phyla indicated in the legend below. 6mA level (%) and total genome size for each representative are shown. (*) *Rhizophagus irregularis* C2 genome size is 159.8 Mb[45]. Source data are provided as a Source Data file.

this lineage-specific observation to a species of a different fungal phylum. Typically associated with transcription repression[47,56,57], the distribution of 5mC pointed to a role in regulation of genome stability. These results showed marked epigenetic differences between two Mucorales species, underscoring the interest in the combined study of distinct species to understand methylome evolution.

### Genomic implications of DNA methylation in Mucorales
DNA methylation differences in *Mucor* and *Phycomyces* prompted us to investigate their roles in these organisms. Like in protists and other

EDF[27,30,58], 6mA sites in *Phycomyces* concentrated downstream of the TSS in methylated adenine clusters (MACs)[30] that range in size from 50 to 500 bp (Fig. 3a, Supplementary Fig. 4). These MACs[30] are less frequent in *Mucor* (39 MACs, 0.2% of genes harbor at least one MAC) than in *Phycomyces* (9843 MACs, 40% of genes harbor at least one MAC) (Supplementary Table 2, Supplementary Data 1). Thus, the distance between adjacent 6mA sites was higher in *Mucor* than in *Phycomyces* (Fig. 3b), in which 92% of 6mA sites were found within clusters versus 0.18% in *Mucor* (Supplementary Table 2). Similar to other basal eukaryotes, 6mA frequency peaked downstream of the TSS, being

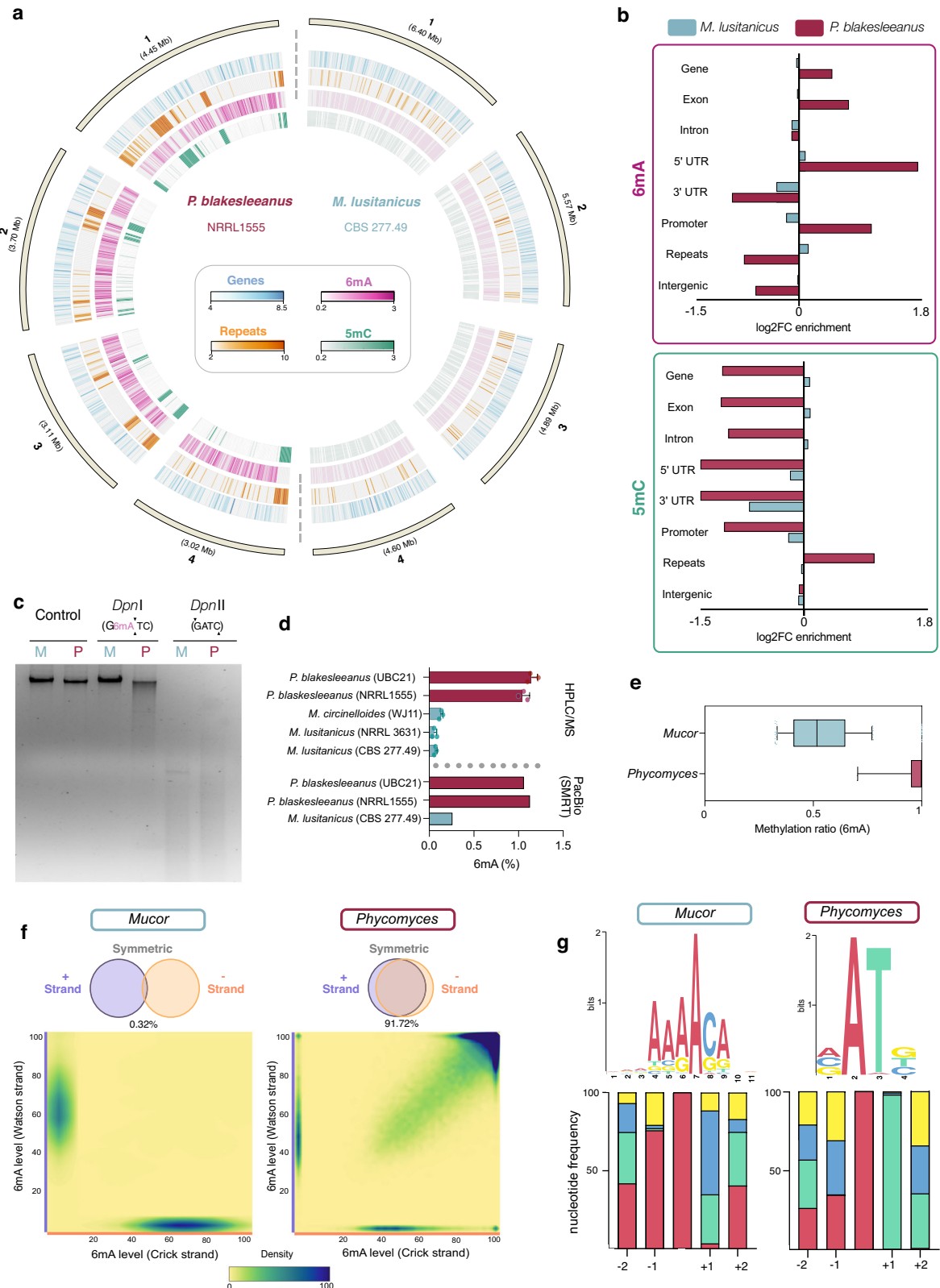

more evident in *Phycomyces* genes (Fig. 3c). This differed from the bimodal 6mA distribution around the TSS described in green algae[28].

RNA-seq analysis of *Mucor* and *Phycomyces* mRNA revealed an association between the presence of 6mA sites and MACs around the TSS (−150 bp to +400 bp) and actively expressed genes in *Phycomyces* (Fig. 3d), consistent with previous observations in other fungi[30], whereas *Mucor* genes lacked this association (Fig. 3d). We also

analyzed the link between 6mA and specific gene functions. Gene Ontology (GO) analysis of 6mA-marked genes revealed enrichment in housekeeping genes involved in functions such as GTPase and ATPase activity, and chromosome organization, similar to other fungi and protists[26,30] (Supplementary Fig. 5). Despite differing in 6mA distribution, there are some common enriched functions between *Mucor* and *Phycomyces*, such as pyrophosphatase activity, while others were

**Fig. 2 | Epigenetic DNA modifications in the *Mucor* and *Phycomyces* genomes.**
**a** View of scaffolds 1–4 from both *Mucor* (right) and *Phycomyces* (left), including gene density tracks (blue), repeats density (orange), 6mA level (pink-purple), and 5mC level (green). **b** Enrichment of 6mA (top) and 5mC (bottom) over genomic features. Results are indicated as log2FC of the enrichment. **c** Inverted agarose gel image of the result of untreated (Control) and genomic DNA of *Mucor* (M) and *Phycomyces* (P) digested with *Dpn*I (cleaves methylated GATC sites), *Dpn*II (cleaves unmethylated GATC sites). This experiment was repeated three times with similar results. **d** HPLC-MS/MS and SMRT quantification of the total 6mA content in the indicated *Mucor* and *Phycomyces* strains. Three biological replicates for each strain were analyzed by HPLC-MS/MS. Data is represented as mean +/- SD. **e** Methylation ratio for 6mA sites in *Mucor* and *Phycomyces*. The ratio was computed as the total methylated reads/total reads for each site. Box bounds represent 25th to 75th percentile with a line at median. Whiskers are drawn from 10th to 90th percentile. (n = 108435 sites for *Mucor* and 355843 for *Phycomyces*) **f** Density plots indicating the methylation ratio for each site in the Watson (*Y*-axis) and Crick (*X*-axis) strands. Notably, by looking at the top right corner, only a few sites in *Mucor* (left) are symmetric (both strands methylated), whereas this proportion is the opposite for *Phycomyces* (right). **g** 6mA motif and nucleotide frequency in each position for methylated sites in *Mucor* (left) and *Phycomyces* (right). Source data are provided as a Source Data file.

species-specific (Supplementary Fig. 5). In *Mucor*, functions related to hydrolase and binding activity resembled those reported in *R. irregularis* (another low-6mA species)[45]. This functional conservation suggested the 6mA importance in low-6mA EDF despite its scarcity. We also found 6mA enrichment in *Phycomyces* genes for 5mC deposition, pointing to a possible interaction and cross-regulation between these DNA modifications (Supplementary Fig. 5).

To further understand the role of 6mA, we analyzed the genome of the *Phycomyces* strain UBC21. UBC21 and NRRL1555 are two *Phycomyces* isolates of different mating type that share a high degree of sequence conservation[59]. The *Phycomyces* UBC21 genome showed a similar 6mA level (1.07%) (Fig. 1, Fig. 2d) and distribution across the genome to NRRL1555 (Supplementary Fig. 6A), with VATB also being the most common motif (Supplementary Fig. 6B). Less conserved regions between strains showed reduced 6mA levels (Supplementary Fig. 7), likely because most of the differences were found in gene-sparse and repeat-rich regions, which were poorly methylated in both genomes. However, we identified genes within these non-conserved regions (Supplementary Table 4) with significantly low 6mA levels (Supplementary Fig. 6C), including genes *sexP* and *sexM*, which determine the mating type. Although *sexM* (in NRRL1555) and *sexP* (UBC21) lacked methylation, the other two components of the sex locus (the triose phosphate transporter and the RNA helicase) harbored a MAC in both strains, suggesting a possible role for 6mA in regulating sex locus genes in *Phycomyces* (Supplementary Fig. 6D). Further studies, analyzing 6mA during sexual interaction and offspring, are needed to assess its role in *Phycomyces* sexual reproduction.

Enrichment of 5mC in repeats prompted us to explore its role in the control of transposable elements (TEs) in *Phycomyces*. Organisms have developed many mechanisms to defend against TE activity, in which small RNAs and chromatin modifications play a central role[60–62]. *Phycomyces* genome analysis showed 5mC enrichment in CG and CHH contexts, but not in the CHG, across both DNA and RNA TEs (Fig. 3e). 5mC spanned the complete TE sequence, peaking at their start and end. RNA TEs displayed a higher methylation than DNA TEs, and LTR TEs were more methylated than non-LTR TEs in both CG and CHH contexts (Fig. 3e).

We also analyzed 5mC distribution in *Phycomyces* protein-coding genes. We divided genes into four quartiles according to their expression levels determined by RNA-seq. Genes in the first quartile (higher expression) were very poorly methylated compared to those in the fourth quartile (lower expression), which showed higher 5mC enrichment in CG and CHH contexts (Fig. 3f). This contrasted with 6mA distribution. Despite being mostly in *Phycomyces* repetitive regions, 5mC marks in gene bodies were also associated with transcription repression, with non-expressed genes ranking among the most methylated (Supplementary Fig. 8).

### Epigenetic and transcriptional regulation during vegetative growth, dimorphic transition, and environmental responses
Given their dynamic nature, epigenetic changes during growth stages are crucial for development[63,64]. We scrutinized 6mA and 5mC, along

with transcriptomic changes, across different growth and morphological stages in both *Mucor* and *Phycomyces*. *Mucor* is a dimorphic fungus, switching between yeast or hyphal growth depending on the environment[65,66]. Additionally, *Mucor* is also known for its high production of industrially relevant lipids like γ-linolenic acid (GLA)[67,68]. Lipid accumulation starts with nitrogen depletion after 24 h of growth (Supplementary Fig. 9A)[69–71]. *Phycomyces* exhibits large fruiting bodies, sporangiophores, attracting research due to their environmental sensing abilities[50,72,73].

We analyzed methylation in *Mucor* (yeast/ germinated yeast, nitrogen +/-, and light/dark) and for *Phycomyces* (light/dark for mycelium and sporangiophore). Overall 6mA levels were similar across all samples. *Phycomyces* 5mC levels were also constant, but *Mucor* showed slight changes with growth conditions and light (Fig. 4a). 6mA motifs and 5mC distribution remained similar across all conditions for both fungi (Supplementary Fig. 10A, C). However, while the number of 6mA/5mC sites did not vary much between high/low expressed genes in *Mucor* for each growth condition/stage, *Phycomyces* showed more 6mA sites in highly expressed genes compared to lower expressed genes, and the opposite trend for 5mC sites across all conditions (Fig. 4c). Consequently, 6mA enrichment downstream of the TSS was clearly noticeable for the higher expressed genes, but absent in the lower and silent genes of *Phycomyces* in all conditions tested (Fig. 4D). Furthermore, a very reduced proportion of the 400 more poorly expressed genes harbored a MAC, whereas around a half of the 400 higher expressed genes contained a MAC across all *Phycomyces* samples (Fig. 4e).

We analyzed gene expression in all these samples and found differentially expressed genes (DEGs) for all comparisons (Fig. 4b, Supplementary Data 2). Comparisons of induction by light yielded lower number of DEGs than the developmental stages (Fig. 4b), similarly to previous observations in *Neurospora crassa*[74–77]. The analysis of the association between DEGs and DNA methylome changes revealed no significant global changes in the methylation ratio for 6mA and 5mC sites for the DEGs in comparisons involving *Mucor* (germinated *vs* yeast, nitrogen- *vs* nitrogen + , and light *vs* dark). In *Phycomyces* comparisons (light *vs* dark in mycelium and sporangiophore—sporangiophore *vs* mycelium growing in the dark or with light exposure), upregulated genes displayed a higher 6mA content than the downregulated genes, while downregulated genes had higher 5mC levels than upregulated genes for all comparisons (Fig. 4f). Additionally, MAC gains and losses were associated with higher and lower expression level, respectively, for both light versus dark and mycelium versus sporangiophore comparisons (Fig. 4g, Supplementary Fig. 10B), indicating a link between 6mA and gene expression in the environmental sensing and the asexual cycle of *Phycomyces*.

Although we did not observe significant global changes in methylation for the *Mucor* comparisons, we conducted a detailed analysis of genes involved in lipid metabolism given their industrial relevance[18,69]. We analyzed 49 genes (Supplementary Table 5) with known roles in lipid metabolism[17,69–71,78–80]. Several of them were differentially expressed when growing in nitrogen-rich versus nitrogen-depleted media, but only 14 out of the 49 genes exhibited differences

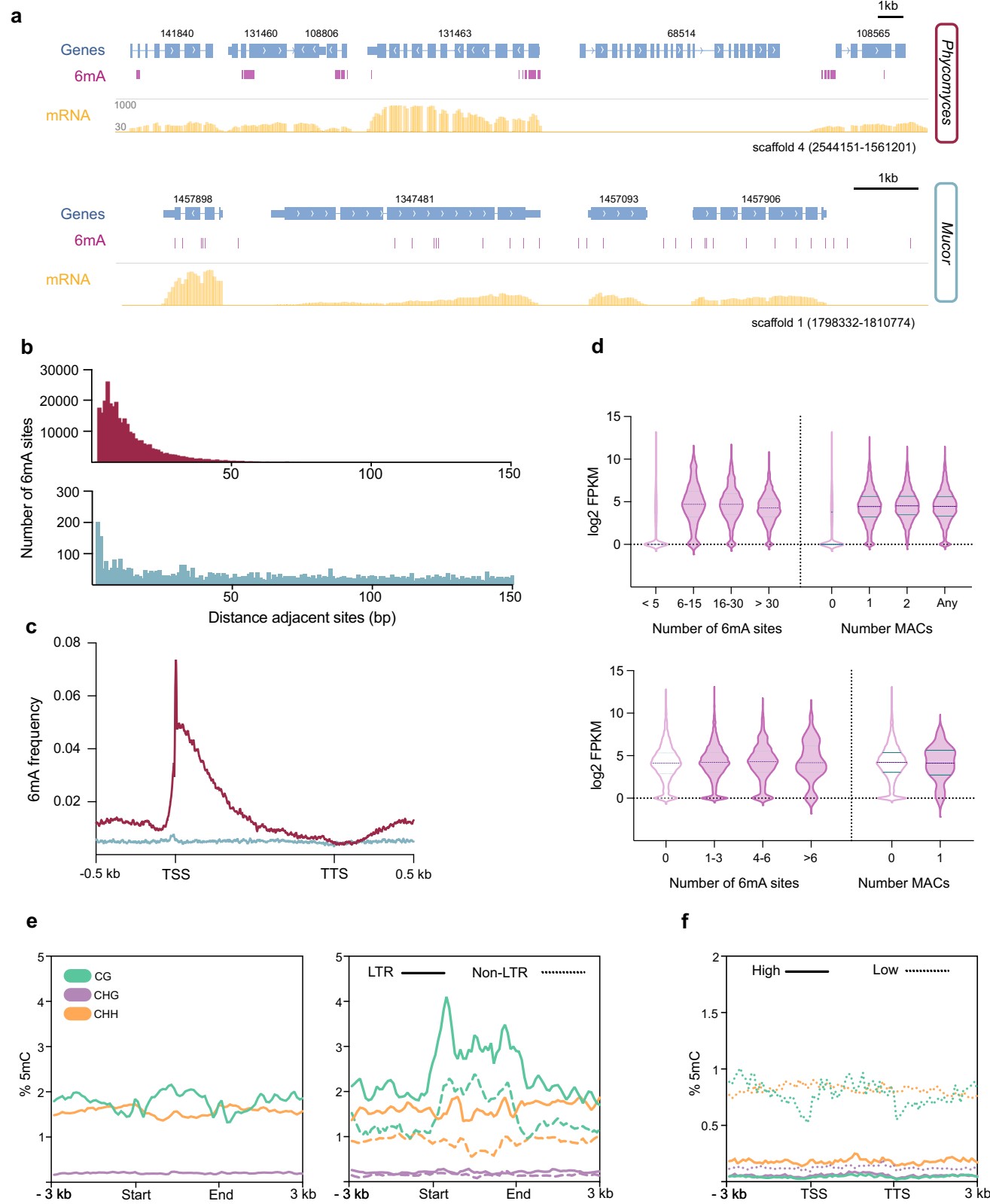

in the number of 6mA sites between both conditions (Supplementary Fig. 9B, C), with those changes restricted to a few sites (Supplementary Fig. 9C).

Light response regulation has been one of the most studied processes in *Mucor and Phycomyces*[50,81–86]. To characterize the light-regulated gene expression and epigenetic remodeling in *Phycomyces*, we generated expression data coupled with 6mA/5mC data for the

"blind" mutant L51 (*madA/madB*)[87]. Expression data revealed no DEGs in light *vs* dark comparisons for mycelium or sporangiophore samples, confirming its blind phenotype (Supplementary Fig. 11D, E). This mutant displayed strong similarity in 6mA and 5mC levels, and 6mA sequence context and distribution at genes to the wild-type strain NRRL1555 (Supplementary Fig. 11A–C, Supplementary Fig. 12D). Additionally, the set of DEGs with differences in the ratio of 6mA sites

**Fig. 3 | Distribution and roles of DNA modifications. a** 6mA marks are frequently clustered in the *Phycomyces* genome (top) and scattered in the *Mucor* genome (bottom). A snapshot of scaffold 4 (*Phycomyces*) and scaffold 1 (*Mucor*) depicts each case. **b** Distance between adjacent 6mA sites in the *Phycomyces* (top) and *Mucor* (bottom) genomes. **c** 6mA frequency profile over *Phycomyces* (purple) and *Mucor* (blue) genes. All genes were scaled to 1.5 kb, divided into equally sized bins and extended 0.5 kb upstream and downstream. 6mA frequency was calculated for each bin. **d** Association between the expression level (FPKM) and 6mA content of *Phycomyces* (top) and *Mucor* (bottom) genes. Number of MACs and 6mA sites were calculated in a window of −150 bp +400 bp from TSS. The log2 FPKM values for each group of genes were plotted using violin plots. Inner lines indicate the media and the interquartile range. **e** Weighted 5mC level at *Phycomyces* DNA transposons (Class II, left) and RNA transposons (Class I, right), distinguishing between LTR retrotransposons (LTR) and non-LTR retrotransposons (Non-LTR). Transposons were scaled to 3 kb and extended 3 kb upstream and downstream the start and end of each element, respectively. Methylation levels are provided for CG, CHG, and CHH contexts and computed on each bin. **f** Weighted methylation level at highly expressed genes and poorly expressed genes. Genes were divided and ranked into fourth quartiles depending on their expression level (FPKM). Genes were scaled to 3 kb and extended 3 kb upstream and downstream the TSS and TTS, respectively. Methylation levels for each context (CG, CHG and CHH) are plotted for the first quartile (High expression) and the fourth (Low expression). Source data are provided as a Source Data file.

in the wild-type NRRL1555 strain (Fig. 4d) did not show differences in the L51 mutant (Supplementary Fig. 12C).

Finally, we selected a curated list of light response genes in these fungi, including blue light photoreceptors and carotenogenic genes[82,84,86,88,89] (Supplementary Table 6). The comparison of the orthologs between *Mucor* and *Phycomyces* genes showed that most of them displayed analogous transcriptomic responses despite the differences in their methylation landscapes (Supplementary Fig. 12A). Examining 6mA variation between dark/light, we observed subtle changes with no specific pattern for any gene (Supplementary Fig. 12B). This provided evidence that, despite the association between 6mA presence and gene activity, this is not the only mechanisms regulating transcriptional responses to environmental changes.

## DNA methylation influences genome defense during fungal development

We also analyzed changes in DNA methylation and TE activity in *Phycomyces* across growth stages and environments. We followed a dual approach, assessing TE activity by expression levels and scrutinizing novel insertions. We found 469 active TEs (FDR < 0.05) in the *Phycomyces* genome, showcasing differences between mycelium and sporangiophore in the wild-type strain (NRRL1555) and the blind mutant (L51) (Supplementary Fig. 13A, C). This suggested less effective TE control in the sporangiophore, potentially creating genome variations for spores' adaptation to new conditions. Comparable active TE numbers were found in NRRL1555 and L51 under all conditions, with light increasing activity in NRRL1555 but not in the blind mutant L51 (Fig. 5a). Specifically, a Class II DNA transposon (Helitron/Maverick) was upregulated in *Phycomyces* sporangiophores in both strains compared to mycelium, while several of Class I LTR retrotransposons were downregulated (Fig. 5a). Interestingly, among the most upregulated TE (log2FC > |4|), Maverick/Polinton1_SM and Tc/Mariner-18 DNA transposons were upregulated in both light and dark in the NRRL1555 and L51 sporangiophores (Supplementary Table 7). In consonance, sporangiophores had more insertions and TE types involved (Fig. 5b). TcMariner TE dominated insertions (85%) across all samples (Supplementary Data 3), validating results from a previous study[90]. Another four TEs (En/Spm, LTR/Pao, Zisupton_6, and LINE CR1-79_HM) generated new insertions across all samples (Supplementary Data 3). Positive correlations between expression and insertions were noted for TcMariner TE ($r = 0.81$), the LTR/Pao ($r = 0.38$), and Zisupton_6 ($r = 0.48$) (Fig. 5c).

When analyzed together, we did not detect a direct correlation between the overall expression levels for all TE copies and their methylation levels within the same growth condition (Supplementary Fig. 13B), likely because both parameters were computed for all copies of the same TE simultaneously. For a more precise assessment, we narrowed the search to TE sets that showed marked transcriptional differences between conditions (Fig. 5d). Highly induced TEs in light-exposed sporangiophores (LS) showed reduced 5mC levels compared to dark conditions (DS) (Fig. 5d). Similarly, transcriptionally active TEs in dark-grown mycelium (DM) exhibited lower 5mC levels compared to those in light-grown mycelium (LM) (Fig. 5d), indicating that 5mC represses TE activity as previously described for protein-coding genes in *Phycomyces* (Fig. 3f, Supplementary Fig. 8). Sorting TEs by 5mC levels, 6mA was frequently absent in those with higher 5mC levels (Supplementary Fig. 13D), supporting their contrasting roles. Consistent with findings in other fungi[91], TE copies within a genome showed higher methylation variation than homologous copies between conditions (Fig. 5e). This heterogeneity in the methylation density on each copy could explain why methylation and expression were not found to be significantly associated when the methylation and expression of all copies were studied together. Still, differential methylation suggested that it plays an important influence on the activity of specific TE during the *Phycomyces* development.

The *Phycomyces* genome encodes DNA cytosine methyltransferases DIM2, DNMT1, and two DNMT2[47]. DIM2 expression remained constant in all conditions, but DNMT1 expression varied, being higher during mycelial growth and lower in sporangiophores. This reduction in DNMT1 expression in sporangiophores could be involved in the aforementioned higher TE activity observed in this stage (Supplementary Fig. 13E).

## MetB is responsible of asymmetric DNA N6-adenine methylation

Prevalence of 6mA in EDF (Fig. 1) motivated us to characterize the responsible machinery. We identified conserved genes in *Mucor* and *Phycomyces* encoding proteins similar to known DNA 6mA-methyltransferases. These included the putative ortholog of N6MT1, a methyltransferase containing a small methyltransferase domain (PF05175) and responsible for 6mA deposition in humans[92] (named *metA*). Additionally, we found an N6-methyltransferase domain-containing protein (PF02384) conserved in EDF[30,47] (named *metB*). We also identified the putative ortholog of the MTA-70 N6- methyltransferase *damt-1* (PF05063) involved in *C. elegans* 6mA deposition[54] (named *metC*). Furthermore, we identified the putative ortholog of the main MTA-70 methyltransferase responsible for 6mA writing in protists (Mta1) (Fig. 6a), which form a complex with another non-active DNA methyltransferase (Mta9) and two DNA-binding proteins (P1 and P2)[27,29,93].

We deleted each gene in *Mucor* (Supplementary Fig. 14) and included in the subsequent analysis a mutant in the *dmt1* gene encoding an ALKBH1 6mA demethylase, which displays developmental defects in *Mucor*[43]. 6mA levels for the wild-type strain MU636 and each mutant using HPLC/MS revealed no significant 6mA reduction in *metA*Δ and *metC*Δ mutants. However, *metB*Δ and the *metA*Δ/*metB*Δ double mutant had a drastic reduction in 6mA levels confirmed by PacBio SMRT sequencing (70-75% reduction) in all conditions (Fig. 6b, c). Noteworthy, *metB* deletion mainly affected asymmetric 6mA, the dominant methylation in *Mucor*. Consequently, the proportion of symmetric 6mA tripled in the *metB*Δ mutant in all conditions, attributable to the reduction of asymmetric 6mA (Fig. 6c), and the characteristic AAA6mACA motif was lost in both *metB*Δ and *metA*Δ/*metB*Δ strains (Fig. 6d), validating this motif for fungal 6mA. On the contrary, we observed no significant changes in 5mC levels and

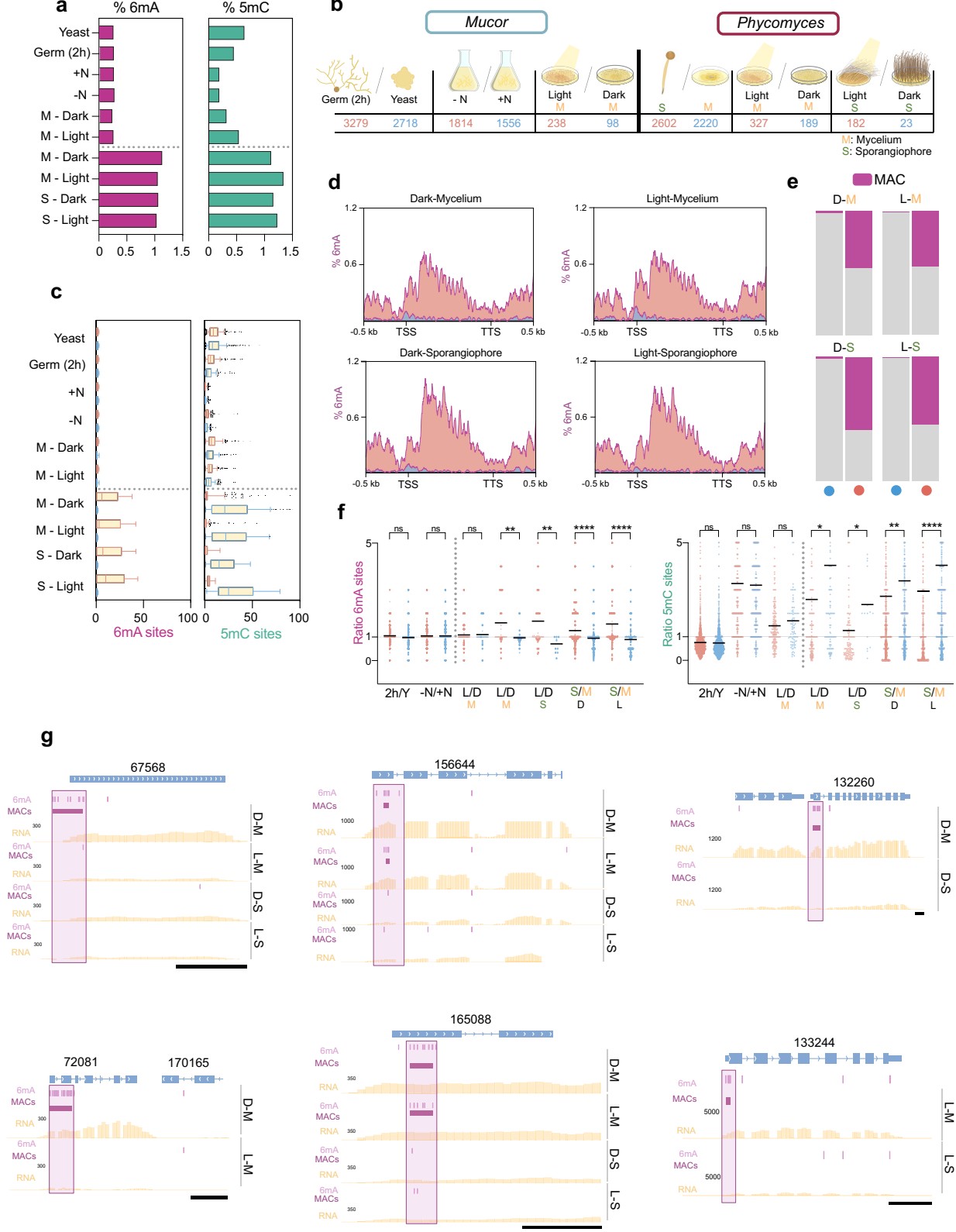

distribution on these mutants (Supplementary Table 2). These results suggested that MetB is the main methyltransferase for asymmetric 6mA deposition in *Mucor* and probably other EDF.

Phenotypic analyses revealed no significant differences in growth and sporulation between the *metAΔ*, *metBΔ*, and *metCΔ* mutants and the wild-type strain. Nevertheless, *metBΔ* was more sensitive to hydroxyurea (Fig. 6e), a DNA synthesis inhibitor (120). Moreover,

*metAΔ* mutants were more virulent in *Galleria mellonella* larvae, while *metBΔ* and *metCΔ* mutants were less virulent (Supplementary Fig. 16G). Additionally, given *Mucor* oleaginous nature, we measured lipid content for all strains. While the *dmt1* mutant exhibited lower lipid content, the *metAΔ* and *metBΔ* mutants displayed a higher proportion, with the double mutant *metAΔ/metBΔ* showing a nearly 100% increase (Supplementary Fig. 15A), making these strains promising candidates

**Fig. 4 | DNA modification in basal fungi development and light sensing.**
**a** Genomic 6mA and 5mC levels for each developmental stage and growth condition. *Mucor*: Yeast, 2 h germination from yeast (Germ 2 h), Nitrogen-rich media ( + N), nitrogen-depleted media (-N) and mycelium (M) in dark and under light exposure. *Phycomyces*: Mycelium (M) and Sporangiophore (S) growing in dark (D) and under light exposure (L). Samples above and below the dotted line correspond to *Mucor* and *Phycomyces*, respectively. **b** Number of DEGs (*P* < 0.05) that are upregulated (red) and downregulated (blue) for each comparison. From left to right: 2 h germination *vs* yeast (2 h/Y) for *Mucor*, nitrogen depleted media *vs* nitrogen rich media (-N/ + N) for *Mucor*, mycelium (M) growing in light *vs* mycelium growing in the dark (L/D) for *Mucor*, sporangiophore (S) *vs* mycelium for *Phycomyces*, mycelium growing in light *vs* mycelium growing in the dark (L/D) for *Phycomyces* and sporangiophore (S) growing in light *vs* sporangiophore growing in the dark (L/D) for *Phycomyces*. **c** Boxplots of the number of 6mA and 5mC sites detected for the top 400 higher (boxes with red borders) and top lower (boxes with blue borders) expressed genes for each condition analyzed. Samples above and below the dotted line correspond to *Mucor* and *Phycomyces*, respectively. Box bounds represent 25th to 75th percentile with a line at median. Whiskers are drawn from 10th to 90th percentile. **d** 6mA distribution over the top 400 more highly expressed genes (red) and lower expressed genes in *Phycomyces*. **e** The proportion of genes (*Phycomyces*) containing a MAC (purple) of the top 400 genes with lower

expression level (left for each sample, blue dot) and the top genes with higher expression levels (right for each sample, red dot). **f** 6mA and 5mC ratio over the upregulated genes (red) and the downregulated genes (blue) in the comparison indicated below the graph, which are the same as those in **b**, except for the *Phycomyces* S/M comparisons where DEGs in dark and light were analyzed independently. A ratio >1 indicates more methylated sites in treatment *vs* control. Samples on the left and on the right the dotted line correspond to *Mucor* and *Phycomyces*, respectively. The mean methylation ratio is indicated with a black line. A two-tailed Welch's test was performed for each comparison (*P*-val 2 h/Y = ns (6mA), ns (5mC); *P*-val -N/ + N = ns (6mA), ns (5mC); *P*-val L/D = ns (6mA), ns (5mC); *P*-val L/D = 0.0017 (6mA), 0.0264 (5mC); *P*-val L/D = 0.0014 (6mA), 0.0306 (5mC); *P*-val S/M < 0.0001 (6mA), 0.0019 (5mC); *P*-val S/M < 0.0001 (6mA), <0.0001 (5mC)), ns = not significant. (n = 2697, 2061, 1400, 1207, 153, 75, 69, 108, 63, 21, 716, 507, 531, and 348 for 6mA sites ratio from left to right comparisons and n = 3151, 2548, 935, 872, 226, 95, 221, 138, 144, 25, 2208, 1601, 1774, and 1330 for 5mC sites ratio from left to right comparisons). **g** Genome snapshot depicting MAC loss/gain (indicated with purple boxes) for DEGs in the different developmental stages (mycelium and sporangiophore) and environmental conditions (dark and light) in *Phycomyces* genes (accession numbers are indicated above each gene). Yellow graphs represent the read counts (FKPM). Purple vertical bar, 6mA; purple horizontal bar, MAC. Scale (black bar) = 500 bp. Source data are provided as a Source Data file.

for commercial lipid production. As nitrogen depletion triggers lipid accumulation, we analyzed 6mA distribution in this condition, revealing differences in the presence of 6mA sites in the gene body of some genes involved in lipid metabolism. Most of these differences were detected in the *metB*Δ and *metA*Δ/*metB*Δ due to the general loss of 6mA on these mutants (Supplementary Fig. 15B). However, global gene expression changes between *metB*Δ and the wild-type strain showed that demethylated genes in the mutant can be both upregulated and downregulated (Fig. 6f). This supports previous findings indicating the absence of an association between asymmetric 6mA and active gene expression, suggesting that asymmetric 6mA might regulate genes differently than symmetric 6mA. Overall, asymmetric 6mA regulates important processes, including DNA replication or repair, virulence and lipid biosynthesis in *Mucor*.

## An MTA-70 methylation complex mediates symmetric 6mA methylation in *Mucor*

Despite the few symmetric 6mA sites, *Mucor* allows the study of the responsible machinery for this type of modification. *Mucor* genome has putative homologs for *mta1*, *mta9* and *p1*, but not for *p2*, suggesting that DNA binding depends only on P1 (Fig. 6g). We measured the 6mA content by HPLC/MS of *mta1*Δ and a new knockout mutant for *p1* (Supplementary Fig. 14A), finding no differences compared to the wild-type strain, which was validated by PacBio SMRT sequencing (Fig. 6h). However, single-site resolution revealed absence of all symmetric 6mA sites and clusters in *mta1*Δ and *p1*Δ strains (Fig. 6h, Supplementary Data 1). The low proportion of symmetric sites in the *Mucor* genome explained the unchanged overall motif (Fig. 6i). Similarly, 5mC levels and distribution did not show changes in any comparison (Supplementary Table 2).

We next analyzed if the loss of the few symmetric 6mA clusters in *mta1*Δ and *p1*Δ affected the expression of the genes associated with them. All genes in the *p1*Δ mutant and all but one in the *mta1*Δ mutant that lost symmetric 6mA clusters were not differentially expressed (Fig. 6j and Supplementary Data 2). We found that two genes involved in DNA damage repair, encoding a DNA helicase PIF1/RRM3 (ID: 1601498) and a G/T mismatch-specific thymine DNA glycosylase (ID: 1452591), were highly upregulated genes in both mutants compared to the wild-type (Supplementary Data 2), suggesting a need for increased repair activity after loss of function of Mta1c and symmetric 6mA. Possibly related to that, both mutants had reduced growth (Fig. 6m, Supplementary Fig. 16B). The *mta1* mutant also showed lower sporulation (Supplementary Fig. 16C) and an increased susceptibility to SDS (Supplementary Fig. 16D, E).

However, mating and virulence were unaffected in the mutants (Supplementary Fig. 16F, H). To confirm the phenotypic cause, we transformed the *mta1*Δ mutant with the wild-type and a mutant allele of *mta1* in which the essential DPPW domain[29] was changed to APPW (Fig. 6k). Only the wild-type allele restored growth (Fig. 6l, m), indicating that the catalytic activity of Mta1 is essential for this phenotype. These results unveiled two distinct DNA 6mA writing machineries in *Mucor*, each specialized in one of the two types of 6mA modifications found in this fungus.

## Asymmetric methylation in EDF could have been acquired through horizontal gene transfer from bacterial endosymbionts

Having elucidated the main components of 6mA deposition machinery in *Mucor*, we aimed to assess their evolutionary trajectory and conservation. Since MTA-70 orthologs are extensively characterized and widely distributed in eukaryotes[94], we narrowed the search to MetB. This methyltransferase belongs to a family of N-6 DNA methylases (N6_Mtase, PF02384) which groups with bacterial restriction methylases from Type I and IC systems. This family appeared limited to prokaryotes and terrestrial EDF from Mucoromycota, Mortierellomycota and Glomeromycota (with some exceptions, Supplementary Data 4, Supplementary Data 5). The HHPred alignment of *Mucor* MetB also revealed a target recognition domain (PF20466), like its homolog-type IIL system methylase Mmel from *Methylophilus methylotrophus* (PDB: 5hr4)[95].

Phylogenetic analyses of bacterial and fungal N6_Mtase methyltransferases showed a well-supported fungal-specific clade, clearly separated from their bacterial counterparts (labeled as clade 1, Fig. 7a, b). Fungal taxa within the fungal clade recapitulated the known taxonomic groups, which is typical for vertically inherited genes. The species composition of the monophyletic fungal clade suggests an ancient horizontal transfer to the common ancestor of *Blastocladiomycota* and terrestrialized fungi, with subsequent loss in most of Zoopagomycota and Dikarya. This is supported by the presence of *metB* in two non-Mucoromycota fungi: *Conidiobolus coronatus* (KXN73575.1) and *Paraphysoderma sedebokerense* (KAI9143530). Noteworthy, bacterial and fungal sequences displayed sequence identity often below 30%, excluding major prokaryotic contamination. Interestingly, a group of Glomeromycota sequences and their endosymbionts sequences was located within a bacterial clade (labeled as clade 2, Fig. 7a, b). The similarity levels between fungal and bacterial proteins (50-80%) and placement of the fungal sequences within a bacterial clade suggests a second horizontal transfer from endosymbiotic bacteria to *Glomeromycota*.

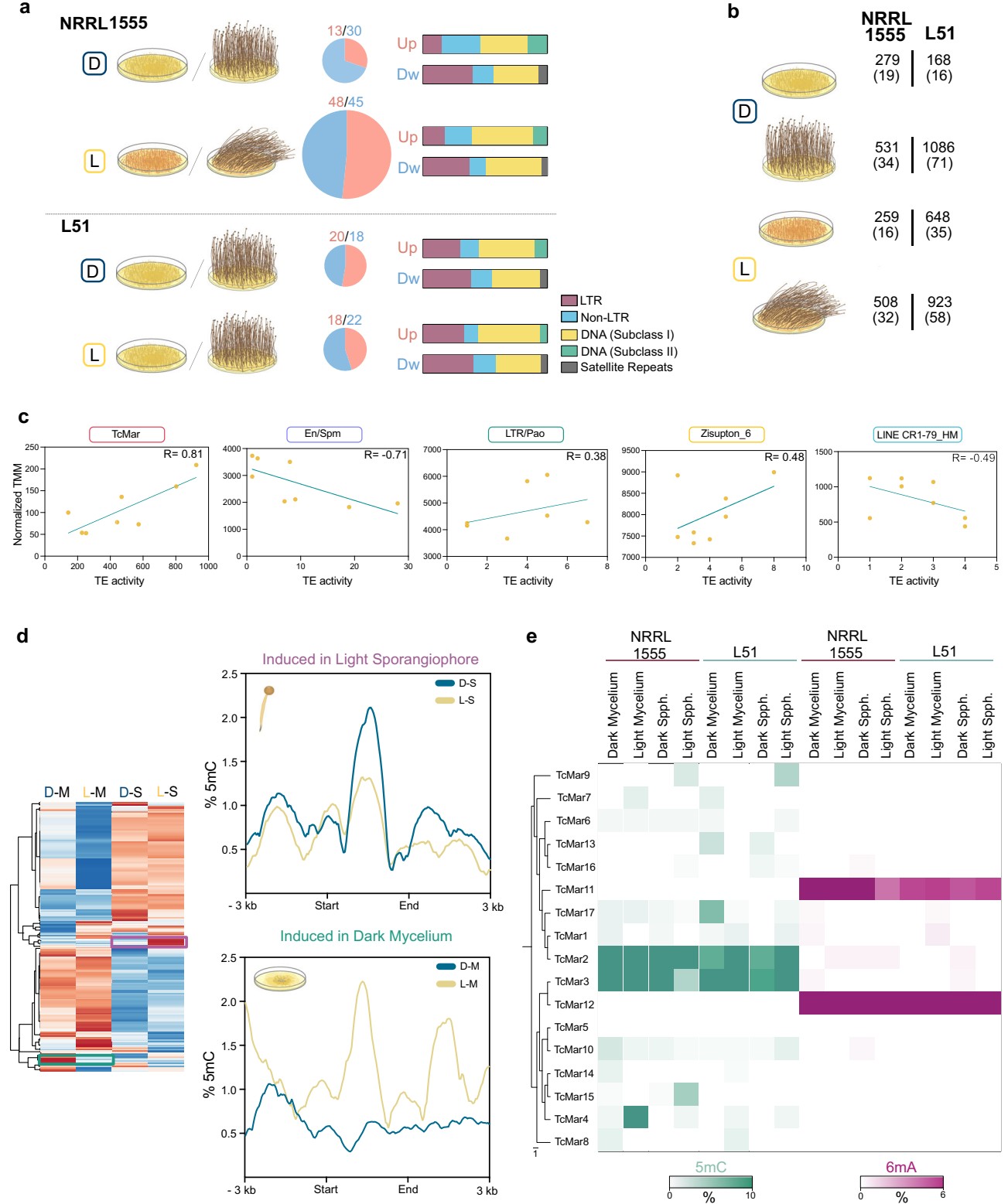

## Discussion

The study of 6mA in eukaryotic DNA remains controversial due to the low levels in some species, particularly in complex eukaryotes with scattered asymmetric 6mA[34,58,96,97]. Conversely, ciliates, green algae, and EDF show high 6mA levels, suggesting they play a biological role[34,58]. Our results reveal that high 6mA varies across EDF, showing marked differences between phyla. Mucoromycota, Mortierellomycota, Kickxellomycota, Zoopagomycota, Neocallimastigomycota and Basidiobolomycota usually display higher 6mA, whereas Chytridiomycota, Monoblepharomycota, Blastocladiomycota, and Glomeromycota show reduced levels, comparable to higher eukaryotes. Interestingly, we also detected significant variability within fungal phylum, highlighting the suitability of EDF for studying the evolutionary trajectories of epigenetic DNA modifications, not only in fungi but also in other eukaryotes.

We use two well-known Mucorales representatives, *Mucor* and *Phycomyces*, to investigate the role of 6mA and its relationship with

**Fig. 5 | Epigenetic regulation of transposable elements of *Phycomyces* during the different growth conditions. a** Proportion of upregulated (red) and down-regulated (blue) TEs in sporangiophore (treatment) and mycelium (control) under dark (D) and light (L) conditions for the indicated strains are shown in pie charts, along with the number of differentially expressed TEs. Proportion of each TE/Repeat class upregulated (red, Up) and downregulated (blue, Dw) is depicted in bar charts. **b** Number of novel insertions detected in each growth condition. The number of different TEs implicated in these insertions is indicated in brackets. **c** Correlation in all assayed conditions between expression level (average TMM) and number of insertions. Pearson's correlation coefficient (R) was computed for each comparison (top right corner of each plot). **d** Hierarchical clustering of TE expression for the wild-type strain NRRL1555 in dark-mycelium (D-M), light-mycelium (L-M), dark-sporangiophore (D-S), and light-sporangiophore (L-S). At the right, the weighted methylation level for TEs most induced in L-S and D-M (purple and green boxes in left graph) are shown. Transposons were scaled to 3 kb and extended 3 kb upstream and downstream the start and end of each element, respectively. **e** Methylation levels (5mC in green and 6mA in red) in the wild-type strain NRRL1555 and blind mutant L51 for the seventeen TcMar transposon copies detected in the NRRL1555 reference genome. Source data are provided as a Source Data file.

5mC in gene expression and response to environmental and developmental cues. These fungi exhibit distinct epigenetic landscapes. *Phycomyces* displays abundant, symmetric, and MAC-distributed 6mA, while *Mucor* has lower, asymmetric, and disperse 6mA. These differences extend to 5mC: high in *Phycomyces* (especially in CG context) and low in *Mucor*. 5mC abundance also varies across the fungal kingdom, with Basidiomycetes and some Ascomycetes showing high levels while Mucoromycota and Zoopagomycota display scarcity[47]. *Phycomyces* deviates from most EDF with high 5mC[47,48,98], adding complexity to its epigenome. *Mucor* could represent the evolution towards the asymmetric 6mA found in Dikarya and higher eukaryotes, whereas *Phycomyces* poses an exception with high 5mC and symmetric 6mA, which could be a model for studying cross-talk between DNA modifications. The finding of genes involved in 5mC methylation among the most 6mA-methylated in enrichment analyses points to a possible cross-regulation between both DNA epigenetic modifications.

The epigenetic landscape characterization across different developmental and environmental conditions, as well as gene expression, revealed that symmetric 6mA and MACs (Supplementary Data 1) in the 5' region of genes are associated with active gene expression in the 6mA-rich *Phycomyces*. This correlation is absent in *Mucor*, which exhibits residual methylated clusters and scattered asymmetric 6mA. Consequently, highly upregulated genes were associated with increasing 6mA contents in *Phycomyces*. While *Phycomyces* and *Mucor* have different dominant 6mA types, some functions are enriched in highly methylated genes of both species. In *Mucor*, methylated genes encode hydrolases and binding proteins, similar to *R. irregularis* (another low-6mA EDF)[45]. This functional conservation suggests that 6mA might have a conserved role even at low levels in EDF.

In *Phycomyces*, similar to other eukaryotes exhibiting high 5mC levels, this epigenetic modification is enriched in repeat-rich regions, including TEs. *Phycomyces* RNA transposons show greater methylation than DNA transposons, especially in the CG and CHH context, spanning the entire TE body with peaks at both ends. Interestingly, 5mC is not restricted to TEs but is also associated with silenced genes and is absent in active genes, consistent with its repressive roles on higher eukaryotes. Analysis of the relation between transposon activity and 5mC levels in different growth conditions (wild-type background, blind background, mycelium, sporangiophore, dark, and light) revealed that highly differentially expressed TEs also display differential 5mC profiles, suggesting that 5mC represses TE transcription and contributes to genome stability. Interestingly, we detected higher variability in methylation between TE copies within a genome than between homologous copies in different samples. This finding aligns with observations[91] in *Marasmius oreades*, where least methylated copies were implicated in new insertions. This agrees with the detection of higher TE insertions at the sporangiophore stage in *Phycomyces*, where there is reduced transcription of the 5mC methyltransferase DNMT1 across all strains and conditions tested.

The methylation machineries for symmetric and asymmetric 6mA in EDF have been characterized by deleting candidate genes in *Mucor*. The rare methyltransferase MetB, found in several EDF lineages, is crucial for asymmetric 6mA as mutants completely lose it, while maintaining symmetric 6mA. Interestingly, *metB* deletion affects the sensitivity to hydroxyurea, lipid biosynthesis, and virulence, indicating a role for asymmetric 6mA in *Mucor* biology. However, further analysis is required to elucidate the underlying mechanisms involved in the cellular processes controlled by asymmetric 6mA. Remarkably, MetB is a part of a prokaryotic system, and phylogenetic analyses support two different HGT from bacteria to EDF (two separate fungal clades embedded in a bacterial tree). The ancient symbiotic interactions between EDF and bacteria could explain the acquisition of this methyltransferase, as some Glomeromycotina contain a second MetB clustered with the sequences of their endosymbionts, which could represent a more recent HGT. Analogously, recent investigations suggest that HGT was responsible for metabolism diversification[99,100] and the acquisition of the RNAi components[101] in some EDF, reinforcing our hypothesis of EDF-bacteria symbiotic associations as the source of the 6mA machinery, particularly the components involved in asymmetric deposition.

Although present at very low levels, *Mucor* symmetric 6mA allowed the identification of an MTA-70 methyltransferase complex involved in methylation of adenines. This complex resembles the ones of ciliates and green algae[27,29], except for having only one DNA binding protein. Deleting genes encoding the catalytic subunit Mta1 or the DNA binding P1 eliminates symmetric 6mA entirely, confirming their functional role in *Mucor*. These mutants grow slowly, produce fewer spores, are susceptible to SDS stress, and show upregulation of genes involved in DNA repair, suggesting a role of symmetric 6mA in DNA damage. Moreover, we confirmed that Mta1 catalytic activity is required for growth and sporulation. We speculate that functional analyses in EDF with high symmetric 6mA levels may reveal further biological implications for 6mA in these fungi. This work opens up avenues for future exploration of the functional significance of 6mA in the realm of EDF biology.

## Methods

### Ethics statement
Experiments and procedures complied with all relevant ethical regulations and were approved by the University of Murcia Biosafety and Ethics Committee of the Universidad de Murcia.

### DNA sequencing, genome assembly and annotation
The DNA for genome sequencing was isolated using the GenElute Plant Genomic DNA Miniprep kit (Sigma-Aldrich), or after a phenol/chloroform extraction as previously described[102]. For the *Mucor* CBS 277.29 genome assembly (https://mycocosm.jgi.doe.gov/Mucci3), PacBio reads (Sequell IIe >10 kb AMPure Bead Size Selection) were combined with Illumina Hi-C reads. The raw sequencing reads were filtered and processed with the JGI QC pipeline to remove artifacts. The Mitochondrial genome was assembled separately from Falcon preassembled reads (preads) (https://github.com/PacificBiosciences/FALCON) using an in-house tool (assemblemito.py), used to filter the reads, and polished with two rounds of RACON (v.1.4.13)[103]. The main genome was assembled using Flye (v2.9-b1768) (https://github.com/fenderglass/Flye) and the resulting Flye assembly was then improved with Hi-C data[104–106]. Generate_site_positions.py from Juicer version 1.6

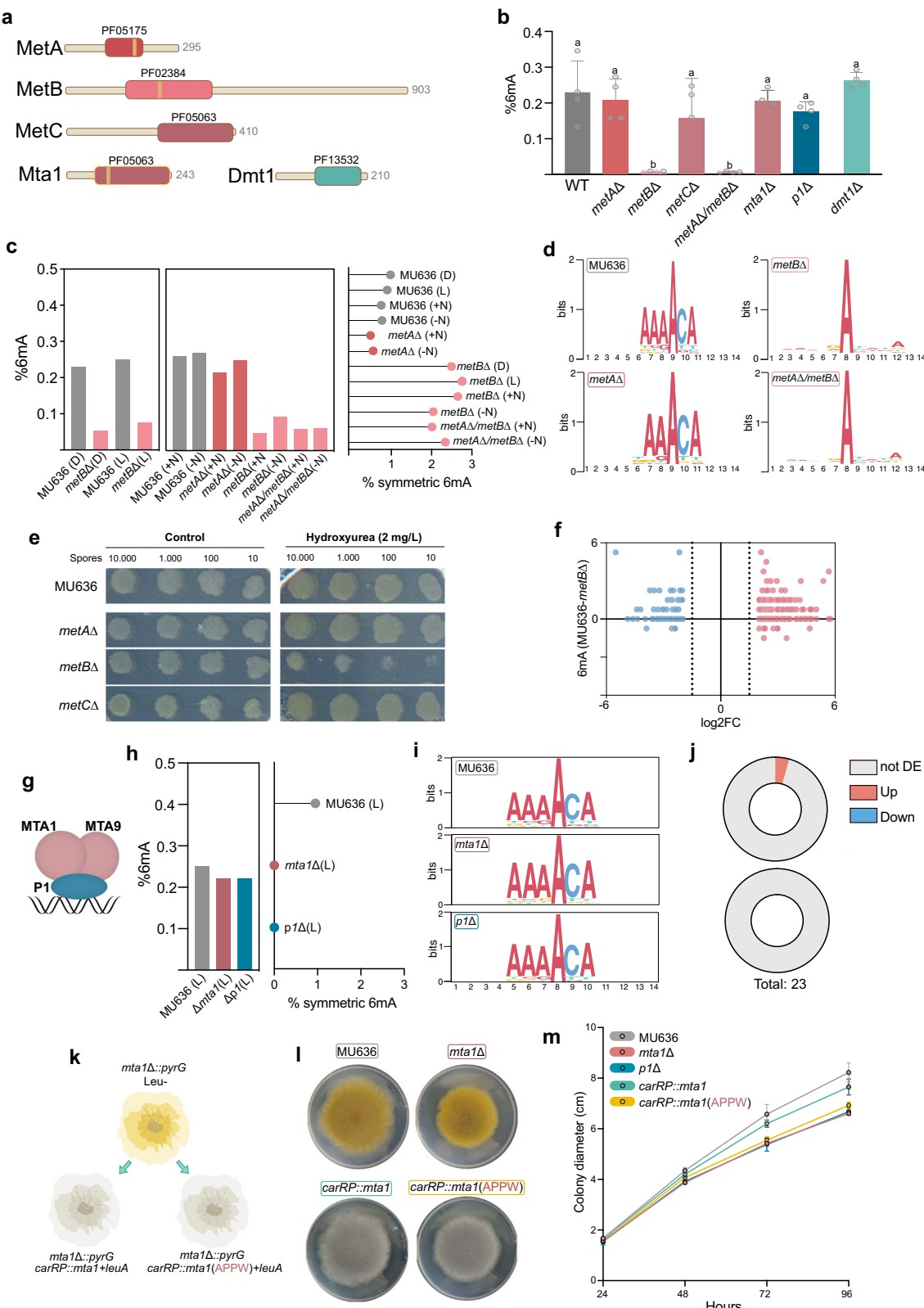

with Arima as the enzyme was used to generate the cut sites. Juicer.sh version 1.6 was used to generate the contact map using 1.5 Gbp of QC-filtered data from the Hi-C library. Manual breaks and joins were made using Juicebox 1.11.08. The final assembly was polished with two rounds of RACON (v1.4.13). *Phycomyces* UBC21 and L51 genome assemblies were generated from PacBio reads (>10 kb Blue Pippin Size Selection). Reads generated were filtered and processed with the JGI

QC pipeline to remove artifacts. Genomes were assembled using Falcon v0.5 and mitochondrial genomes were assembled as described above. Given their close relationship with *Phycomyces* NRRL1555, gene models were manually refined to maximize details on gene function. Telomeric regions were identified at the end of each contig with EMBOSS fuzznuc[107] using the putative telomeric repeat search pattern of (TTAGGG)2 on each strand. *Mucor* centromeric regions were

**Fig. 6 | The methylation machinery involved in dual 6mA distribution in early-diverging fungi. a** Schematic diagrams for the candidate proteins of *Mucor* characterized in this study. PFAM domain and amino acid length are indicated for each protein. **b** 6mA levels measured by HPLC/MS for the wild-type strain (MU636) and the knock-out mutants in *metA*, *metB*, *metC*, *mta1*, *p1*, *dmt1* and the double mutant in *metA* and *metB* genes. Different letters indicate statistically significant differences, while the identical letters denote no significant differences, calculated using one-way ANOVA (*P* < 0.001, Tukey test, Supplementary data 7). Four biological replicates were analyzed for each strain. Data is represented as mean +/- SD. **c** 6mA levels detected with PacBio for the wild-type strain and *metA*Δ, *metB*Δ, *and metA*Δ*/metB*Δ mutants growing in the dark (D), light (L), nitrogen-rich media (N) or nitrogen-depleted media (-N). At the right, the percent of symmetric 6mA sites for each strain and condition is indicated. **d** Motifs for 6mA sites in the wild-type strain (MU636) and in the *metA*Δ, *metB*Δ, *and metA*Δ*/metB*Δ mutant strains. Data from all strains growing in the same conditions (nitrogen-rich media) was used. **e** Growth of 10.000, 1000, 100, and 10 spores growing with and without hydroxyurea (2 mg/L) at 48 h post-inoculation. A reduced growth was observed in the *metB*Δ mutant strain. **f** Differential methylation sites for all DEGs in the Δ*metB* compared to the wild-type strain. The x-axis indicates the log2FC for each gene and the y-axis the difference in the number of 6mA sites. **g** Schematic representation of the

methylation complex involved in symmetric 6mA methylation in *Mucor*. **h** 6mA levels were measured by HPLC/MS for the wild-type strain (MU636) and the knock-out mutants in *mta1* (MU1335) and *p1* (MU1357) in the light (L). At the right, the percent of symmetric 6mA sites for each strain and condition is indicated, no symmetric sites were detected in the MU1335 and MU1357 mutants. **i** Motifs of 6mA sites for the wild type (MU636), *mta1*Δ, and *p1*Δ strains. All motifs were calculated from data derived from all three strains growing under light conditions. **j** Expression of genes that had lost symmetric 6mA cluster in the *mta1*Δ (top chart) or *p1*Δ (bottom chart) strains in comparison to the control strain MU636. Not DE, not differentially expressed; Up, upregulated in the mutant; Down, downregulated in the mutant. **k** Diagram of the experimental procedure followed for complementation of the MU1335 strain. One strain was complemented with the wild-type version of the *mta1* (*carRP::mta1*) and another strain was generated by using a mutated version of *mta1* (*carRP::mta1*(APPW)). Both complemented strains display a white color as a result of *carRP locus* disruption, which is used for mutant screening. **l** Images showing colony growth for MU636, the *mta1* knockout mutant MU1335, and representative complemented strains. **m** Colony diameter (cm) measured at 24, 48, 72, and 96 h post-inoculation for each strain as indicated in the colored legend. Measures were taken from four biological replicates. Source data are provided as a Source Data file.

retrieved[108] and transferred to the new genome assembly. Conservation between *Phycomyces* NRRL1555 and UBC21 was computed in Vista Browser (https://genome.lbl.gov/vista/index.shtml). Regions with conservation greater than or equal to 70% were considered as conserved regions.

## PacBio Isoseq
From 1ug of total RNA as input, full-length cDNA was synthesized using template-switching technology with SMARTer PCR cDNA Synthesis kit (Clontech). The first-strand cDNA was amplified with PrimeSTAR GL DNA Polymerase (Clontech) using template switching oligos to make double-stranded. Double-stranded cDNA was purified with AMPure PB beads and either non-size selected, selected for 2-10 kb or 4 kb cutoff by BluePippin (Sage Sciences) based on the target transcriptome size. The amplified cDNA was end-repaired and ligated with blunt-end PacBio sequencing adapters using SMRTbell Template Prep Kit 1.0. The ligated products were treated by exonuclease to remove unligated products and purified by AMPure PB beads. PacBio Sequencing primer was then annealed to the SMRTbell template library and sequencing polymerase was bound to them using Sequel II Binding kit 2.0. The prepared SMRTbell template libraries were then sequenced on a Pacific Biosciences's Sequel II sequencer using v2 sequencing primer, 8 M v1 SMRT cells, and Version 2.0 sequencing chemistry with 1×1800 sequencing movie run times.

## 6mA SMRT detection and data analysis
6mA base modifications were detected with the PacBio SMRT analysis platform (cromwell.workflows.pb_basemods). 10 ug of genomic DNA was sheared to >10 kb using Covaris g-Tubes. The sheared DNA was treated with exonuclease to remove single-stranded ends and DNA damage repair mix followed by end repair and ligation of blunt adapters using SMRTbell Template Prep Kit 1.0 (Pacific Biosciences). The library was purified with AMPure PB beads and size selected with BluePippin (Sage Science) at >10 kb cutoff size. PacBio Sequencing primer was then annealed to the SMRTbell template library and sequencing polymerase was bound to them using Sequel Binding kit 3.0. The prepared SMRTbell template libraries were then sequenced on a Pacific Biosciences Sequel sequencer using v3 sequencing primer, 1 M v3 SMRT cells, and Version 3 sequencing chemistry with 1×600 sequencing movie run times. Raw reads were filtered using SFilter, to remove short reads and reads derived from sequencing adapters. Filtered reads were aligned to the reference genome for *Mucor* CBS 277.49 v3 or *Phycomyces* NRRL1555 v2 using BLASR (1.5.0)[109]. Modified sites were identified through kinetic analysis[110]. For

each species, one individual and growth condition was used to detect 6mA distribution[30]. 6mA sites were filtered with a minimum of 15x coverage and > 25mQv in all sequencing experiments, except MU1301(+ N) and MU1306 (D) which overall yielded a lower average coverage and were filtered with a minimum of 5x coverage and > 20mQv. Methylated sites were clustered following a previously established procedure[30]. Briefly, motifs are used to define MACs, where modification density within clusters and relative distance between methylated motifs are calculated. Here, the AT motif was used. Relative distance refers to how many unmodified motifs are permitted between modified motifs; lower relative distances are more efficient as they allow fewer unmethylated AT dinucleotides per cluster. The relationship between these two features (modification density and efficiency) is then used to determine optimal clustering. For each sample, relative distances from 1 to 40 were analyzed and the distance with the best score (density*efficiency) was used for genome-wide MAC identification. The sequence context of methylated sites was obtained by extracting ± 6 nucleotides of each site and analyzing those sequences by MEME-ChIP (v5.5.5)[111]. 6mA levels in previously PacBio sequenced EDF (Table 1) were obtained following the same approach from SMRT reads. Enrichment over genomic features was calculated as the ratio between the observed and expected 6mA. Genomic sequences not classified either as genes (exon, intron, 5'UTR, and 3'UTR), repeats, or promoters (600 bp upstream the TSS) were considered intergenic regions. 6mA profile graphs were generated by computing 6mA frequency over equal size bins using deepTools (v3.1)[112]. Symmetry density plots were generated using the methylation ratio for each site and plotted using a scatter of points with kernel density estimations in two dimensions with ggplot2[113]. Gene Ontology enrichment analysis for 6mA methylated genes in *Phycomyces* was done by selecting genes harboring at least one MAC (with at least 10 methylated sites) from 150 nucleotides upstream to 400 nucleotides downstream of the TSS and by selecting the top 400 more densely methylated genes in *Mucor*. GO annotations were retrieved from the *Mucor* CBS 277.49 v3 *and Phycomyces* NRRL1555 v2 annotations. A *P*-value of 0.05 was used as cutoff for the identification of enriched terms (Fisher's exact test).

## 6mA determination by HPLC-MS/MS, dot blot, and isoschizomer digestion assays
Genomic 6mA abundance was measured by HPLC-MS/MS following a previously developed and tested procedure with EDF[43]. Genomic DNA (gDNA) (three biological replicates) from *Mucor* and *Phycomyces* strains was digested into nucleosides with a combination of 1.5 U of

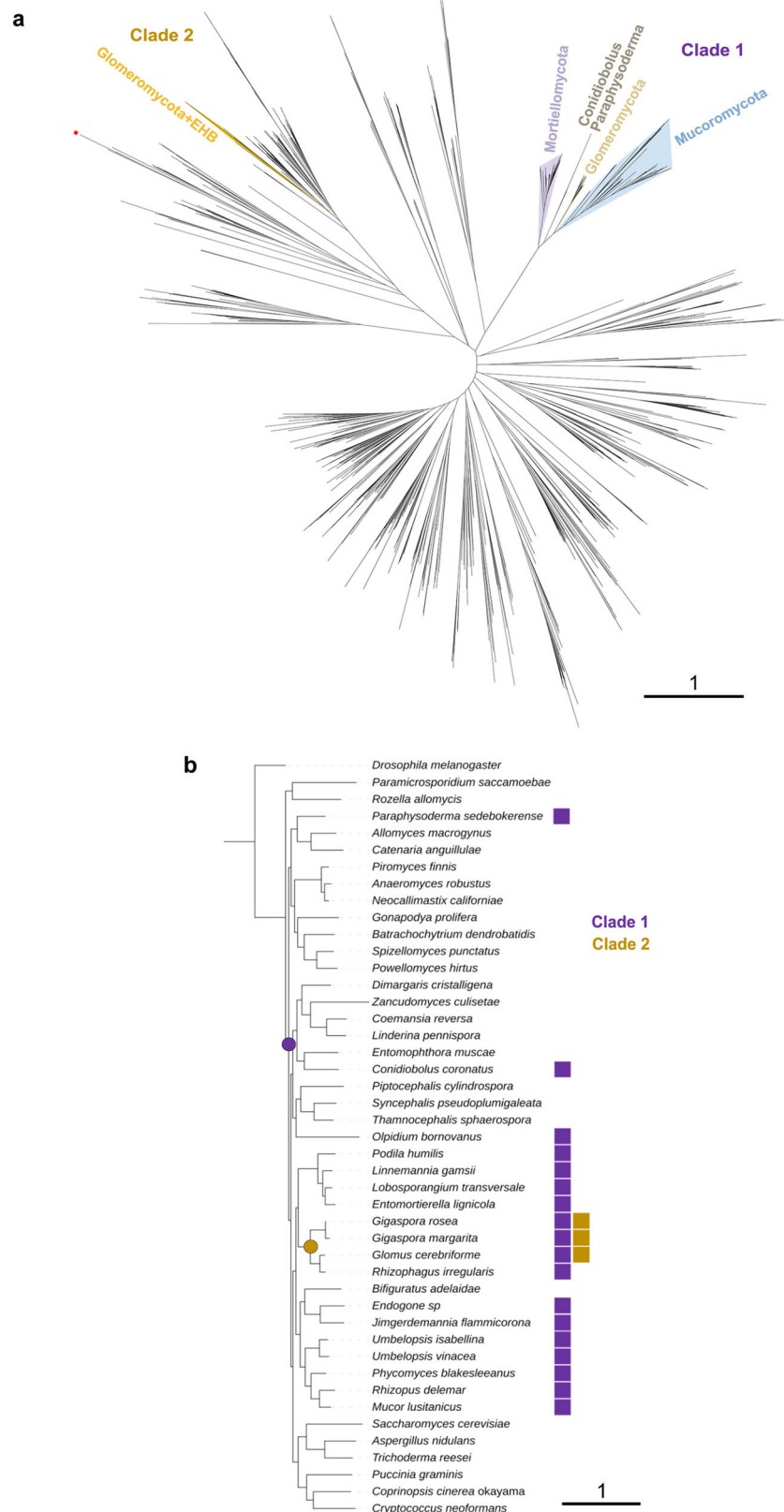

**Fig. 7 | Phylogenetic analyses of MetB homologs. a** Maximum likelihood phylogenetic tree of fungal and bacterial N6_Mtase homologs. Fungal clades are marked in color. Red square denotes a *Batrachochytrium dendrobatidis* sequence (EGF76048.1) found in a *Bacillus*-like clade, which is likely a bacterial contaminant since its sequence comes from an unplaced genomic scaffold encoding other bacterial sequences. In consequence, this sequence was removed from tree reconciliation. **b** Species tree showing the occurrence of N6_Mtase homologs from clade 1 and clade 2 with potential HGT events marked as dots.

DNAse I overnight at 42 °C (ThermoFisher) and 0.001 U of Phosphodiesterase I from *Crotalus adamanteus* venom (Merck, Darmstadt, Germany). The digested gDNA was treated with 1 U of alkaline phosphatase at 37 °C for 2 h and diluted two-fold with nuclease-free water. Single nucleotides were analyzed using an HPLC-MS system consisting of an Agilent 1290 Infinity II Series HPLC (Agilent Technologies, Santa Clara, CA, USA) equipped with an Automated Multisampler module and a High-Speed BinaryPump, and connected to an Agilent 6550 Q-TOF Mass Spectrometer (Agilent Technologies, Santa Clara, CA, USA) using an Agilent Jet Stream Dual electrospray (AJS-Dual ESI) in the positive mode. Experimental parameters for HPLC and Q-TOF were set in MassHunter Workstation Data Acquisition software (Agilent Technologies, Rev. B.08.00). The nucleosides were quantified using the nucleoside to base ion mass transition of 252.1091 > 136.0638 *m/z* for dA (C10H13N5O3) and 266.1248 > 150.0812 *m/z* for 6mA (C11H15N5O3). The 6mA/dA ratio was calculated based on the concentration of each nucleoside. For each experiment, we analyzed three biological replicates and obtained the mean and SD. A dot blot assay for 6mA detection was executed as follows: DNA was quantified using Qubit 4.0 and denaturalized for 5 min at 95 °C. Drops (2 µl) containing 100, 50, and 10 ng were placed in an Amersham Hybond-N+ membrane (GE Healthcare, RPN203B). DNA was crosslinked with 1200 microjoules (50 s) UV, and the membrane was washed with PBS + 0.002% Tween-20 to remove unbound DNA. The membrane was blocked for 1 h with 5% non-fat milk and subsequently incubated with a primary anti-6mA antibody (1:1000) (Synaptic Systems, #202003) overnight at 4 °C. This specific antibody is widely used in dot blot experiments to detect 6mA in DNA (https://sysy.com/product/202003). After washing with PBS + 0.002% Tween-20, HRP labeled secondary antibody was added (1:5000). The Amersham ECL detection kit (RPN2108) was used for visualization. For differential DNA digestion assays, 5U of *Dpn*I, 5U of *Dpn*II, or 0.5 µl of ddH₂O (Untreated) were added to 1 µg of genomic DNA and incubated for 4 h at 37 °C. The digestion products were separated in a 1.5% agarose gel at 100 *V* for 3 h. A representative, uncropped image of this type of gel is provided in the Source Data file.

## RNA extraction and expression data analysis

*Mucor* and *Phycomyces* total RNA was isolated by using the RNeasy Mini kit (Qiagen) and treated with DNAse (Sigma, On-Column DNaseI treatment set). Three biological replicates were analyzed for each expression experiment. Plate-based RNA sample prep was performed on the PerkinElmer Sciclone NGS robotic liquid handling system using Illumina's TruSeq Stranded mRNA HT sample prep kit utilizing poly-A selection of mRNA following the protocol outlined by Illumina in their user guide: support.illumina.com/sequencing/sequencing_kits/truseq-stranded-mrna.html, and with the following conditions: total RNA starting material was 1 ug per sample and 8 cycles of PCR was used for library amplification.

Both types of libraries were quantified using KAPA Biosystem's next-generation sequencing library qPCR kit (Roche) and run on a Roche LightCycler 480 real-time PCR instrument. The quantified library was then multiplexed with other libraries, and the pool of libraries was then prepared for sequencing on the Illumina NovaSeq 6000 sequencing platform using NovaSeq XP v1.5 reagent kits, S4 flow cell, following a 2×150 indexed run recipe. Raw fastq file reads were filtered and trimmed using the JGI QC pipeline. BBDuk (https://sourceforge.net/projects/bbmap/) was used to detect artifacts by kmer matching (kmer = 25), allowing 1 mismatch, and the detected artifacts were trimmed from the 3' end. RNA spike-in reads, PhiX reads, and reads containing any Ns were removed. Quality trimming was performed using the phred trimming method set at Q6. Finally, following trimming, reads under the length threshold were removed (minimum length 25 nucleotides or 1/3 of the original read length). For expression analysis, filtered reads from each library were aligned to the reference genome using HISAT2 (v2.2.0)[114]. Raw gene counts were

calculated with featureCounts[115]. These raw counts were used to to determine DEGs with DESeq2 (v1.30.0)[116] (*P*-value < 0.05 as cutoff). Counts files were also used to generate FPKM and TPM information. Strand-specific coverage bigWigs were generated using deepTools (v3.1)[112]. Expression data were also used for genome annotation. De novo transcriptome assembly was performed using Trinity (v2.11.0)[117] (--normalize_reads --jaccard_clip).

## Bisulfite-seq and 5mC data analysis

Genomic DNA (1 µg) was sheared around 500 bp using the LE220-plus Focused-ultrasonicator (Covaris) and subjected to end repair, A-tailing, and ligation of Methylated Indexed Illumina Adapter (IDT). The EZ DNA Methylation-Lightening Kit (Zymo Research) was used for bisulfite conversion of unmethylated cytosine to uracil and clean-up of adapter-ligated DNA. Libraries were amplified using 10 cycles of PCR and purified using AMPure Purification (Beckman Coulter). Purified libraries were sequenced with Illumina HiSeq-2500. Raw fastq file reads were filtered and trimmed using the JGI QC pipeline. Using BBDuk (https://sourceforge.net/projects/bbmap/), raw reads were evaluated for artifact sequence by kmer matching (kmer = 23), allowing 1 mismatch and the detected artifacts were trimmed from the 3' end. Quality trimming was performed using the phred trimming method set at Q6. Reads under the length threshold were removed (minimum length 25 bases or 1/3 of the original read length). Finally, one nucleotide to the right of the reads was trimmed to prevent the creation of completely contained read pairs. Reads were mapped to the *Mucor* and *Phycomyces* reference genomes with bowtie1 (seed length 50, maximum mismatches in the seed set to 1, and maximum insert size set to 1000)[118]. For each species, one individual and growth condition was used to detect 5mC distribution. Reads were deduplicated and reads that mapped to multiple locations were removed. Methylation calling was performed with Bismark (v0.24)[119]. Weighted methylation levels were calculated by dividing the total number of methylated reads by the total number of unmethylated plus unmethylated reads for each sequence context[120] and methylated sites were filtered using a minimum of 5x coverage and 10% methylation ratio. Profile metaplots over genes and transposable elements were generated by computing weighted methylation levels over equal size bins using deepTools (v3.1)[112]. Motif and genomic features enrichment analysis was conducted as described for 6mA. Correlations between expression data and bisulfite data were obtained with MethGET (v2.0.4)[121].

## Transposon analysis

Repeats and transposable element sequences were identified using RepeatModeler2[122] (v2.0.4) (-LTRstruct). The resulting output was fed into RepeatMasker (v4.1.4) (http://www.repeatmasker.org) to detect every repeat in the genome (-s). Lower scoring matches whose domain partly (<80%) includes the domain of another match were removed from the final repeat list. To generate a more curated TE list, the RepeatModeler library was translated using EMBOSS transeq[107] (v5.0.0) in all possible six frames with -clean option enabled and scanned using InterProScan[123]. Only those consensus sequences that reported hits related to TE activity were retained[124,125] and a new input library was generated to run RepeatMasker with a strict cutoff value (-s -cutoff 4500 -no_is, -nolow -norna). For the study of active TEs in *Phycomyces,* we used two different approaches. First, we studied whether specific TEs were expressed in our samples. We used the ExplorATE (v0.1.0)[126] tool using the TE annotation obtained with RepeatMasker and RNA-seq data. In total, 469 active TEs were detected. Second, we studied whether the 469 active TEs had generated new insertion sites in the different samples with respect to the *Phycomyces* reference genome. For this purpose, we used the tool Tefinder[127] (v1.0.1). For TcMar phylogenetic classification, only full-length and DDE endonuclease domain-containing (PF13358) copies were kept. Sequences were aligned using MAFFT[128] (v7.508) and a maximum

likelihood tree was generated with IQ-TREE[129] (v2.2.2) with 1000 ultrafast bootstrap replicates.

## Phylogenetic analyses

Proteomes of the 63 representative species included in Fig. 1 were retrieved from the Joint Genome Institute (JGI) MycoCosm genome portal[51], including those generated for this paper, which are publicly available on their specific portals. Using Orthofinder (v. 2.5.4)[130] 335 orthogroups with a minimum of 85% of species with single copy orthologs in any of them were detected and aligned using MAFFT (v.7.471)[128]. Alignments were concatenated, trimmed, and used to infer the species tree with RAxML (raxmlGUI v 2.0.10)[131,132] with PROTGAM-MAWAGF substitution model and 1000 bootstrap replicates. For MetB analyses, 183 fungal proteomes (Supplementary Data 4) were scanned with pfam_scan.pl against the PFAM database to identify all fungal N-6 DNA Methylase (PF02384) homologs. Fungal sequences were inspected and clustered with cd-hit (70% sequence identity)[133], then used as queries in subsequent blastp[134] searches against the NR database (threshold E-value 0.001)[135]. Collected sequences were clustered in CLANS[136]. Domain architecture was predicted using InterProScan[123] and HHPred via MPI-Toolkit[137]. Sequence alignments were calculated with Mafft v.7[128] (localpair, maxiterate 100) and trimmed with TrimAl[138] (gappyout). Phylogenetic trees were calculated in IQTree2[139] (-B 1000 -alrt 1000). A phylogenomic tree was built with OrthoFinder[130] (msa, mcl, fasttree) for a set of 45 taxa with *Drosophila melanogaster* as an outgroup. Phylogenetic trees were visualized in iTOL[140]. For sequence accessions of fungal *metB* homologs and count of *metB* homologs in 183 fungi, see Supplementary Data 4.

## Fungal growth conditions and phenotypic characterization

For light and dark growth experiments, 1000 spores from *Mucor* were inoculated on solid YPG medium (pH 4.5) and grown for 18 h in the dark. The plates used for dark growth conditions were incubated for another 18 h in the dark, whereas those used for growth under light exposure were transferred to an incubator with white light exposure for 18 h. For yeast/hyphal transition samples, $1\times10^6$ spores/ml were inoculated in a 50 mL tube filled with liquid to the lid with YPG medium (pH 4.5) and saturated with $CO_2$. Tubes were incubated in the dark overnight without shaking to induce yeast growth. Hyphal growth was induced by transferring those cultures to a 500 mL flask and by incubating with oxygenation for 2 h at 250 rpm in a rotary shaker. For nitrogen depletion/lipid metabolism samples, $1\times10^6$ spores were inoculated in nitrogen-rich medium KR[141] (3.3 g/L ammonium tartrate, 7 g/L KH$_2$PO$_4$, 2 g/L Na$_2$HPO$_4$, 1.5 g/L yeast extract, 0.007 g/L CaCl$_2$, 1.5 g/L MgSO$_4$, 8 mg/L FeCl$_3$, 1 mg/L ZnSO$_4$, 0.1 mg/L CuSO$_4$, 0.1 mg/L Co(NO$_3$)$_2$, 0.1 mg/L MnSO$_4$, and 30 g/L glucose). After 12 h, mycelium was filtered through a 70 μm cell strainer and half of the total mycelium was reinoculated in fresh nitrogen-rich medium KR for another 12 h, while the other half was transferred to a nitrogen-depleted and glucose-rich KR media (without ammonium tartrate, replacing yeast extract with 0.5 g/L of yeast nitrogen base w/o amino acids and 50 g/L glucose) and incubated for 12 h. All these experiments were carried out in 500 mL flask with baffles and incubated at 250 rpm in a rotary shaker. For radial growth assays, 1000 spores were inoculated in the center of a YPG medium plate (pH 4.5) and colony diameter was measured at 24, 48, 72, and 96 h post-inoculation. Experiments were performed by triplicate. For the phenotypic characterization of sexual interaction, plates of YPD medium were inoculated with 500 spores of MU1335 (-), MU636 (-) and NRRL 3631 (+) and incubated for 15 days in the dark (see Supplementary Table 8 for strain description). The production of zygospores (a darker region in the interaction region) was visualized. For sporulation assays, the total number of spores in a 1 cm2 chunk of agar were quantified (one per plate, 9 plates total) after 72 h of growth on YPG (pH 4.5). For stress assays with SDS, 150 spores were inoculated in MMC pH 3.2 plates supplemented with 0.002% SDS. The

survival ratio was calculated with the total number of colonies in control and SDS plates. Experiments were performed by triplicate. For the hydroxyurea (HU) stress test, 10.000, 1000, 100, and 10 spores were inoculated in MMC pH 3.2 plates supplemented with 2 mg/L of HU. Ammonium assimilation was determined by the indophenol method[142]. The total lipid content was quantified following previously established procedures for *Mucor*[79]. Starting from 20 mg of biomass, lipids were extracted with chloroform/methanol (2:1, v/v) and methylated with 10% (v/v) methanolic HCl at 60 °C for 3 h. All *Mucor* cultures were grown at 26 °C. For DNA or RNA isolation from *Phycomyces* mycelia, cultures were inoculated on minimal medium agar plates and grown at 22 °C in the dark for 2 days, exposed to light (1 W/m² blue light) during 30 min or kept in the dark as control. To isolate DNA or RNA from sporangiophores, cultures were initiated as described above and grown for 2 days in the dark at 22 °C, after which mycelium was exposed to light (1 W/m² blue light) for 2 min to induce sporangiophore initiation and returned to the dark for another day. Sporangiophores were then either exposed to light (1 W/m² blue light) for 30 min or kept in the dark, removed, and stored at −80 °C before DNA or RNA extraction.

## Mutant strain generation

*Mucor strain* MU402[37] (Ura⁻, Leu⁻) was used as the recipient strain to generate the following single gene deletion mutant strains (Supplementary Table 8): MU1301 (*metA*−ID:1544229), MU1306 (*metB*−ID:1597305), MU1307 (*metC*−ID:1475563), MU1335 (*mta1*−ID:1550399) and MU1357 (*p1*−ID:1569123) following a previously established procedure[143]. Briefly, a recombination fragment was constructed using *pyrG* as a selectable marker flanked by 1 kb sequences upstream and downstream for each of the targeted *loci* (Supplementary Data 6). Transformants were selected in MMC medium[144] ((pH 3.2) supplemented with 1 mg/L of niacin and 1 mg/L of thiamine) and homokaryotic mutants were selected after 5-7 cycles of growth in MMC medium. Gene deletion and homokaryosis were confirmed either by PCR or Southern blot. For Southern blot validations, DNA probes that allowed the discrimination between wild-type and mutant *loci* (Supplementary Data 6) were labeled using α-³²P dCTP using Read-To-Go Labeling beads (GE Healthcare Life Sciences) and hybridized to a total of 1 μg of DNA digested with the specified restriction enzymes, separated by gel electrophoresis and transferred to an Amersham Hybond XL membrane (GE Healthcare Life Sciences). For PCR validation, primers flanking the targeted locus that allowed to discriminate between the wild-type and mutant loci were used (Supplementary Data 6). For the double knockout mutant (MU1310), a recombination fragment carrying the *leuA* selectable marker was used instead of *pyrG*. The strain MU1301 (Ura⁺, Leu⁻) was used as the recipient strain to knockout the *metB* (ID: 1597305) gene. In this case, transformants were selected in YNB medium[145] (pH 3.2) supplemented with 1 mg/L of niacin and 1 mg/L of thiamine) and homokaryotic mutants and selection was performed as described before. Genetic complementation of *mta1* with the functional and mutated version of the gene was performed as follows: the wild-type mta1 ORF and a mutagenized non-functional version of the gene (DPPW to APPW) were PCR amplified (Supplementary Data 6) and cloned into pMAT1476[146] using *Nhe*I and *Sac*II restriction sites. The newly generated plasmids (pMAT1826 for the wild-type version of *mta1* and pMAT1827 for the mutagenized version) carried the cloned ORFs (including the native promoter and terminator) and the *leuA* selectable marker flanked by the 1 kb regions *carRP* homology regions. Plasmids were digested with *Nhe*I and *Sac*II to release the template for homologous recombination and protoplasts of the MU1335 strain were transformed as described above. Disruption of *carRP* allows for convenient screening for white colonies in YNB medium (pH 3.2. Supplemented with 1 mg/L of niacin and 1 mg/L of thiamine), white colonies should be homokaryotic strains that carry the desired *mta1* integration. Disruption of the *carRP* locus and the

absence of wild-type nuclei were confirmed by color colony screening and PCR.

## Determination of virulence potential in *G. mellonella*

Spores from strains grown on YPG were harvested in sterile spore buffer (0.9% w/v NaCl and 0,01% v/v Tween20), filtered through a sterile 40 μm cell strainer to minimize number of hyphal elements, washed by centrifugation in sterile IPS (insect physiological saline: 150 mM NaCl, 5 mM KCl, 10 mM EDTA, and 30 mM sodium citrate in 0.1 M Tris–HCl, pH 6.9) three times and filtered through a 10 μm filter. The resulting suspension was then adjusted to the preferred inoculum concentration ($5 \times 10^7$ spores/ml). Sixth instar larvae of *G. mellonella* (SAGIP, Italy), weighing 0.4–0.5 g, were selected for experimental use. Larvae, in groups of thirty, were injected through the last pro-leg into the hemocoel with $10^6$ spores in a volume of 20 μl, as described in Maurer et al.[147], and incubated at 30 °C in the dark. Untouched larvae and larvae injected with sterile IPS served as controls. Survival was determined every 24 h over 6 days. This experiment was repeated at least three times with similar results.

## Statistical information

Phenotypic characterization results were expressed as mean ± S.E. The data were analyzed with IBM SPSS Statistic software for Mac [IBM Corp (2014) Version 23.0.; https://www.ibm.com/SPSS-Statistics/]. Data normality was analyzed using the Shapiro-Wilk normality test with a significance level (alpha) of 0.05. An ANOVA of a single factor was used to determine statistically significant differences between groups of data with normal distribution, assuming a significance level of 95% ($P$-value < 0.05), followed by Tukey's' HSD post-hoc test. Direct comparisons between the expression levels of methylated and unmethylated genes were analyzed using Welch's test. Statistical differences in *G. mellonella* survival analyses were assessed using Log-Rank (Mantel-Cox) and the Gehan-Breslow-Wilcoxon test. Pearson correlation factors, $P$-values cut-off for Fisher exact tests, and Welch's test are indicated in the respective figure legends.

## Reporting summary

Further information on research design is available in the Nature Portfolio Reporting Summary linked to this article.

## Data availability

The raw sequence data that support the findings of this study have been deposited in the Sequence Read Archive (SRA) under the accession numbers: SRP496792, SRP496793, SRP496795, SRP496796, SRP496797, SRP496798, SRP496799, SRP496800, SRP496801, SRP496802, SRP496803, SRP496804, SRP496805, SRP496806, SRP496807, SRP496808, SRP496809, SRP496810, SRP496811, SRP496812, SRP496813, SRP496814, SRP496815, SRP496816, SRP496817, SRP496818, SRP496819, SRP496820, SRP496821, SRP496822, SRP496823, SRP496824, SRP496825, SRP496830, SRP496831, SRP496832, SRP496833, SRP497169, SRP497170, SRP497171, SRP497172. All genomes assembled as part of this study are available through DDBJ/ENA/GenBank accession numbers JBDQZJ000000000 (*Mucor* CBS 277.49), JBDFSA000000000 (*Phycomyces* UBC21), and JBCLYO000000000 (*Phycomyces* L51), as well as via MycoCosm through the following links: https://mycocosm.jgi.doe.gov/mycocosm/PhyblU21_2 (*Phycomyces* UBC21), https://mycocosm.jgi.doe.gov/mycocosm/Phybl_L51_1 (*Phycomyces* L51), and https://mycocosm.jgi.doe.gov/mycocosm/Mucci3 (*Mucor* CBS277.49). Source data are provided with this paper.

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

## Acknowledgements

This research was funded the MCIN/AEI/ 10.13039/501100011033 by "ERDF A way of making Europe," by the "European Union" (grant PID2021-124674NB-I00 to F.E.N. and V.G. and PID2021-128001OB-I00 to L.M.C. and D.C.), Fundación Séneca-Agencia de Ciencia y Tecnología de la Región de Murcia (20897/PI/18 and 21969/PI/22 to V.G.), and the Andalusian Government research project P20-00622 funded by the "Consejería de Transformación Económica, Industria, Conocimiento y Universidades" to L.M.C. The work (10.46936/10.25585/60001127) conducted by the US Department of Energy Joint Genome Institute (https://ror.org/04xm1d337), a DOE Office of Science User Facility, was supported by the Office of Science of the US Department of Energy under Contract No. DE-AC02-05CH11231. We would like to thank Joseph Heitman (Duke University School of Medicine), Joseph W. Spatafora (Oregon State University), and Alexander Idnurm (University of Melbourne) for their initial intellectual support for the project. Lastly, we would like to thank Andrey Novichkov for assisting with some of the 6mA analyses at JGI and the 1000 Fungal Genomes Project for being a valuable source of genomic data.

## Author contributions

C.L. conducted most of the *Mucor* experiments, prepared the *Mucor* material for sequencing, performed bioinformatic analysis of the results, prepared the figures and tables, designed and coordinated the project, and wrote the manuscript with significant input from F.E.N., S.J.M., A.M., D.C., M.F., T.G, U.B., Y.S., I.V.G., F.E.N., L.M.C., and V.G. S.J.M. participated in the assembly of genomes and bioinformatic analyses. M.O.C participated in the generation of *Mucor* knockout mutants and analyzed their phenotype. G.G. collaborated on the bioinformatic analyses. A.M. analyzed the phylogeny of *metB*. M.C.-L. produced the *Phycomyces* material for sequencing. R.R. participated in genome assembly. A.L. conducted transcriptomic sequencing. J.G. conducted bisulfite sequencing and Iso-Seq. H.H. and M.A. contributed to library construction and sequencing. V.N. managed the project. D.L.G. participated in the generation of *Mucor* knockout mutants and analyzed their virulence. U.B. analyzed the virulence of the *Mucor* knockout mutants. J.Y. and Y.S. analyzed lipid accumulation. E.N. managed the project and provided materials. D.C., M.F., and T.G. analyzed the results. F.E.N. participated in the generation of *Mucor* knockout mutants, analyzed the results, and supervised the study. L.M.C. analyzed the results, designed, and supervised the study. I.V.G. supervised and coordinated the project. V.G. analyzed the results and designed, supervised, and coordinated the project.

## Competing interests

The authors declare that they have not competing interests.

## Additional information

[1]Departamento de Genética y Microbiología, Facultad de Biología, Universidad de Murcia, Murcia, Spain. [2]U.S. Department of Energy Joint Genome Institute, Lawrence Berkeley National Laboratory, Berkeley, CA 94720, USA. [3]Department of Agricultural Biology, Colorado State University, Fort Collins, CO 80523, USA. [4]Environmental Genomics and Systems Biology Division, Lawrence Berkeley National Laboratory, Berkeley, CA 94720, USA. [5]Institute of Biochemistry and Biophysics, Polish Academy of Sciences, Pawinskiego 5A, 02-106 Warsaw, Poland. [6]Departamento de Genética, Facultad de Biología, Universidad de Sevilla, Sevilla, Spain. [7]Institute of Hygiene and Medical Microbiology, Medical University of Innsbruck, Innsbruck, Austria. [8]College of Food Science and Engineering, Lingnan Normal University, Zhanjiang 524048, China. [9]Colin Ratledge Center for Microbial Lipids, School of Agricultural Engineering and Food Science, Shandong University of Technology, Zibo 255049, China. [10]Department of Biochemistry and Biophysics, Oregon State University, Corvallis, OR 97331, USA. [11]Barcelona Supercomputing Centre (BSC-CNS), Plaça Eusebi Güell, 1-3, 08034 Barcelona, Spain. [12]Institute for Research in Biomedicine (IRB Barcelona), The Barcelona Institute of Science and Technology, Baldiri Reixac, 10, 08028 Barcelona, Spain. [13]Catalan Institution for Research and Advanced Studies (ICREA), Barcelona, Spain. [14]Centro de Investigación Biomédica en Red de Enfermedades Infecciosas (CIBERINFEC), Barcelona, Spain. [15]Department of Plant and Microbial Biology, University of California Berkeley, Berkeley, CA 94720, USA. ✉e-mail: corrochano@us.es; fnicolas@um.es; vgarre@um.es

