## [Peer Review File · Nature Communications]

Symmetric and asymmetric DNA N6-adenine methylation regulates different biological responses in MucoralesREVIEWER COMMENTS

Reviewer #1 (Remarks to the Author):

In the manuscript, "Symmetric and asymmetric DNA 6-methyladenine regulates different biological responses in early-diverging fungi," Lax and co-author compared DNA 6-methyladenine and DNA 5-methylcytosine patterns with gene expression analyses in response to developmental and environmental changes. They focused on two species of early diverging fungi with contrasting 6mA and 5mC content to investigate the role(s) and interplay of these two DNA modifications using a powerful combination of high-throughput genomic techniques and functional experiments. To go beyond the simple correlation analyses that are standard for this type of genomic study, they generated knock-out mutants for both 6mA and 5mC candidate methyltransferase genes. The authors examined the effects of these deletions on the mutant epigenomes and transcriptomes. Following a pioneering paper (Mondo et al., 2017) describing the existence, relative abundance, and phylogenetic distribution of 6mA in early diverging fungi, this new work represents a major contribution to the study of the role of this previously overlooked epigenetic mark. Overall, the study is well done and the paper is well written. The experimental design is sound and the results fully support the conclusions drawn. Subject to some modifications and clarifications, which are listed below, I think that the scope and quality of this manuscript is suitable for publication in Nature Communications.

Major comments:

- Perhaps I've missed this information, but it seems to me that the number of replicates performed for each of the experiments in this study is not clearly stated, either in the text, the figure legend, or the Materials and Methods section. In particular, it is important to mention the technical and/or biological replicates that have been carried out and to provide the results of correlation analyses where appropriate. For example, with respect to Figure 1A, it is not indicated whether the detection of 6mA and 5mC for *Mucor* and *Phycomyces* was performed on several independent individuals from each accession (CBS 277.49 and NRRL1555, respectively) or on a single individual, whether reads from different runs were pooled before mapping to the genome, etc. Similarly, plot 1D shows SD bars, but we do not know from how many samples they were measured. This is true for all experiments described in this manuscript. Therefore, information on the number and type of replicates should be added.
- In general, the figure legends are so short that some information is missing. In several cases, an acronym that appears on the figure is not described in the legend (e.g., see Figure 5D, where there is an explanation of what D vs. L and M vs. S stand for, but what about W?) Sometimes, as in Figure 4F, the legend is even incomplete and thus does not properly describe the figure. In Figure 7A, some fungal clades are not found on the tree (Glomeromycota and Conidiobolus + paraphysoma). Therefore, the figure legends need to be completed appropriately.
- Regarding the increased virulence and lipid content of the mutant strains (821-833), I wonder if it's appropriate to focus so much on them in this article. These are certainly two very interesting phenotypes, but the authors don't provide any explanation as to why virulence is increased and why lipids are overproduced in these mutant strains. It seems to me that this could be the subject of further studies and that in the context of this manuscript it is either too much or not enough.

Minor comments:

- Line 469: How many representatives of Zoopagomycota and Monoblepharomycota does this represent?
- Line 569: Does this mean that there is one MAC per gene? In this case it would mean that all genes are epigenetically marked to be expressed by default?
- Line 590: Why is this enzymatic activity considered common and enriched? Is this demonstrated by this study or previous work (citation required in this case)?
- Line 593: Is functional conservation of 6mA profiles an overstatement as only two species are compared, one of which shows so few 6mA?
- Line 778: The SDs show that there is no significant difference in growth. Therefore, the phrase

"...with a subtle increase during light growth, ..." does not seem appropriate.

- Line 816-817: If the 6mA modification is not necessary for development, is it possible that it is important for fitness? Would it be possible to test this hypothesis?

- Line 835: This observation suggests that the regulation of expression by 6mA is both direct and indirect. Therefore, before concluding that there is no association between asymmetric 6mA and active gene expression, it might be interesting to investigate whether the AAAACA motif, which should be present at the TSS of the down-regulated set of genes, would be absent in the up-regulated one.

- Line 988-989: This argument seems weak because it would be impossible to genetically transform mammalian cells whose genomes have a complex repertoire of epigenetic modifications.

Reviewer #2 (Remarks to the Author):

In this study, authors have investigated the distribution of 6mA among the genomes of early-diverged fungi and narrowed their investigation targets mainly for two species. They found that the species with high 6mA levels distributed in gene-rich regions are associated with symmetric deposition hosting positively correlated expressed genes. In contrast, species with low 6mA levels exhibit asymmetric 6mA modifications with no obvious regulation of gene expression association. This also coincides with the enzymes in charge of methylation of DNA. The study is very interesting to have a large variety of contents. However, I have some concerns about the methodology, design and the main results interpretation.

First, the main part of the study is actually about the comparison of two species of fungi, not about general EDF. It is nice that the authors' design has focused on specific two species with deeper focusing on the differences between two species, but this cannot be generalized. In the general EDF analysis, authors already have shown that the 6mA patterns vary a lot among EDF without generic pattern and even there is high degree of differences among the different species of same phylum. However, at the same time, authors have picked two fungal species and compare the differences. Obviously, authors should be very careful to generalize the finding. For me the study is a mixture of two different contents, but feels like enumeration due to the results that cannot be generalized. The title and abstract are very misleading compared to the actual contents authors have presented. I would strongly recommend authors to change the title and abstract to be specific to two species of fungi, instead of making too much generalization and emphasizing for the whole EDF.

Second, for the materials and methods, there are some points to be clarified and improved. Especially for the RNAseq data analysis and normalization. If I follow the order of sentences, authors used both FeatureCounts and DESeq2. Although FeatureCounts can just generate raw transcripts abundance, authors have specified that FPKM was calculated. Then in the next sentence authors used DESeq2 where FPKM should not be taken for the analysis. DESeq2's input must be the raw counts, as DESeq2 will normalize the transcript abundance by own way which is not FPKM. Normally, in transcriptomics, you don't use FPKM at all if you are analysing data by DESeq2. Therefore, in the case I concern if the whole transcriptome analysis data and related interpretations of this study are based on wrong normalization and consequent outcome. If authors did mistake, then the transcriptome analysis should be re-conducted before the revision. Another part of M&M is about the statistical analysis. Authors used parametric ANOVA as a default, and then used post-hoc depending on the heteroscedasticity. By the way, the test authors have applied is Tukey's HSD, not Tucky. Anyway, if the normality is not tested and confirmed, it is incorrect to use the parametric ANOVA. There was no indication of normality in the part. If some of the data do not have normality, then authors should have applied non-parametric approaches to test the hypotheses. This part should be properly improved. Also, lastly, it seems there are some data showed heteroscedasticity as authors did Games-Howell post-hoc. However, for the T-test, authors simply used Student's T test. Weren't there any unequal variances of data? If so, authors should have used Welch's T test.

Third, for the 6mA occurrence and gene expression tests and the interpretation, it would be better to categorize the different occurring points of 6mA, before challenge the corresponding gene expression. It has been known that not only the symmetric or asymmetric 6mA affects the gene

expression, but also the methylated sites (upstream, promoter, gene body, downstream) can differently affect the gene expression. Have more detailed expression information with each case will make authors interpretation more convincing for the readers. At the same time, since the 5mC occurrence was also investigated, authors can also provide more holistic overview of the methylation and gene expression relationships.

Forth, Although authors have focused on the finding the homologs of methyl transferases associated with 6mA, the observed functions of homologs sometime are different than known homologs. Indeed, some DNA methyl transferases can work for both 5mC and 6mA. Therefore, it is difficult to really affirm that the observed difference in gene expression and phenotypic changes are solely from the changes in 6mA. Authors should clarify there was no change 5mC in each case.

Fifth, about the HGT, authors stated that the MetB was only conserved in certain lineages listed in the manuscript, but with exceptions. In actual data, there are other basal lineages having the enzyme. Clearly, the enzyme occurred from the basal lineages of Fungal kingdom, or even more ancient ancestors in the tree of life. At the same time, the information based on ProFam database is mainly biased by bacteria due to functional characterization. In this case, can author really say that the enzyme is originated "from" bacteria "to" fungi? What makes authors to only consider one direction, or the enzymes have a same origin of HGT? For me, it is very difficult to make HGT claim without proper analysis. Proper phylogenetic analysis should be done for the claim. Does the gene have different codon usage? Or Does synteny of the genome have any indication of HGT? I think the claim should be carefully addressed with clear data.

For the discussion part, I think the discussion can be carefully re-addressed after the additional data analyses and checking.

I just have few minor comments in discussion.

L1081-1082 – It seems EDF-bacteria HGT was also reported in other fungi. PMID: 29329439

Reviewer #3 (Remarks to the Author):

The manuscript entitled "Symmetric and asymmetric DNA 6-methyladenine regulates different biological responses in early-diverging fungi" by Lax et al. impressively unravels the presence and function of an underrepresented and understudied epigenetic modification, DNA adenine methylation (6mA), in an equally understudied group of fungi, the early-diverging fungi (EDF). EDF were identified to have comparatively higher levels of 6mA than Dikarya or higher eukaryotes, making them promising model organisms to study this modification.

The authors have chosen two different EDF species for their comparative epigenetic analysis under different physiological conditions – a very powerful experimental setup – as these are characterized by different DNA methylation profiles: While *Phycomyces* (*P. blakesleeianus* NRRL1555) has 1.13% of adenines methylated, *Mucor* (*M. lusitanicus* CBS 277.49) has only 0.26% of adenines methylated, as quantified by Pacbio SMRT sequencing. *Phycomyces* adenines are symmetrically methylated, while *Mucor* adenines are asymmetrically methylated. Additionally, *Phycomyces* has high levels of 5mC, while *Mucor* again has low levels of this DNA modification. In *Phycomyces*, there is a good correlation between 5mC and gene repression, as well as 6mA and gene activation (likely due to the symmetric modification), while this is not the case in *Mucor*.

The authors then go on to delete 6mA DNA methyltransferase candidate genes, and thereby identify and characterize the responsible enzymes for both the symmetric and asymmetric modification, suggesting heterologous gene transfer for the rare asymmetric 6mA methyltransferase MetB.

Overall, this manuscript is well thought out, well researched and written, and was a pleasure to read. The following comments are intended to make some of the conclusions possibly more clear and stronger:

- Line 545: Does the expression of DNA methyltransferase genes match the observed (different) levels of 5mC and 6mA in the two analyzed EDF species?
- Line 595: Did you notice this in *Mucor*, and could you check whether in your deletion mutants the expression level of the respective 5mC-specific genes is affected?
- Line 810: Going in line with the previous comment - symmetric 6mA is increased in the knockout of *metB*, which catalyzes the asymmetric modification. Would this regulation be performed on a transcriptional level?
- Line 822: How do you explain the reduced virulence of the knockout of *metC*, and even increased virulence of the knockout of *metA*, while they are not contributing to 6mA levels? What do you suggest for their function?
- Line 866: Unfortunately, this contradicts your narrative that symmetric 6mA is correlated with active gene expression. Do you have any hypothesis why *Mucor* might not fit your hypothesis?

Minor comments:

- Line 521: Please make a connection to symmetric/asymmetric modifications, which is only mentioned in the next paragraph.
- Line 531: When analyzing
- Line 543: Is this relevant, is this complex conserved in EDF?
- Typo in legend of figure 4: 5 6mC

REVIEWER COMMENTS

REVIEWER #1:

In the manuscript, "Symmetric and asymmetric DNA 6-methyladenine regulates different biological responses in early-diverging fungi," Lax and co-author compared DNA 6-methyladenine and DNA 5-methylcytosine patterns with gene expression analyses in response to developmental and environmental changes. They focused on two species of early diverging fungi with contrasting 6mA and 5mC content to investigate the role(s) and interplay of these two DNA modifications using a powerful combination of high-throughput genomic techniques and functional experiments. To go beyond the simple correlation analyses that are standard for this type of genomic study, they generated knock-out mutants for both 6mA and 5mC candidate methyltransferase genes. The authors examined the effects of these deletions on the mutant epigenomes and transcriptomes. Following a pioneering paper (Mondo et al., 2017) describing the existence, relative abundance, and phylogenetic distribution of 6mA in early diverging fungi, this new work represents a major contribution to the study of the role of this previously overlooked epigenetic mark.

Overall, the study is well done, and the paper is well written. The experimental design is sound and the results fully support the conclusions drawn. Subject to some modifications and clarifications, which are listed below, I think that the scope and quality of this manuscript is suitable for publication in Nature Communications.

Response: We acknowledge the reviewer's recognition that our work goes beyond establishing correlations and represents a significant contribution to the study of 6mA in early diverging fungi. We appreciate all the comments raised, as they have contributed to improving the revised manuscript. Please note that we have condensed the text to meet the journal's requirements, while preserving the core information.

Major comments:

Comment #1: Perhaps I've missed this information, but it seems to me that the number of replicates performed for each of the experiments in this study is not clearly stated, either in the text, the figure legend, or the Materials and Methods section. In particular, it is important to mention the technical and/or biological replicates that have been carried out and to provide the results of correlation analyses where appropriate. For example, with respect to Figure 1A, it is not indicated whether the detection of 6mA and 5mC for *Mucor* and *Phycomyces* was performed on several independent individuals from each accession (CBS 277.49 and NRRL1555, respectively) or on a single individual, whether reads from different runs were pooled before mapping to the genome, etc. Similarly, plot 1D shows SD bars, but we do not know from how many samples they were measured. This is true for all experiments described in this manuscript. Therefore, information on the number and type of replicates should be added.

Response: Thank you for bringing to our attention the absence of indications regarding the number and type of replicates in some experiments. This information has been added to either the Method section or Figure legends for all experiments, including those mentioned by the reviewer (line 631, 646, 826-827, 1280, and 1352). Most experiments have been conducted with at least three biological replicates, except for 6mA and 5mC profiling, based on our previous experience in which we found 99% overlap using different technologies (Figure S2 from Mondo et al., 2017. Nat Genet. 49:964-968. doi: 10.1038/ng.3859).

Comment #2: In general, the figure legends are so short that some information is missing. In several cases, an acronym that appears on the figure is not described in the legend (e.g., see Figure 5D, where there is an explanation of what D vs. L and M vs. S stand for, but what about W?) Sometimes, as in Figure 4F, the legend is even incomplete and thus does not properly describe the figure. In Figure 7A, some fungal clades are not found on the tree (Glomeromycota and Conidiobolus + paraphysoma). Therefore, the figure legends need to be completed appropriately.

Response: In addition to indicating the number of replicates in response to comment #1, we have also reviewed all figure legends to provide complete information and correct any mistakes to enhance the clarity of the figures. Specifically, we added necessary information to the Figure 5D legend, and extended the legends for Figures 2, 3, and 4. Additionally, we annotated Figure 7A clades for enhanced readability and reconstructed the phylogenomic tree by including *Paraphysoderma*, which was previously missing in Figure 7B. Consequently, we have updated the genome list included in Supplementary File 1.

Comment #3: Regarding the increased virulence and lipid content of the mutant strains (821-833), I wonder if it's appropriate to focus so much on them in this article. These are certainly two very interesting phenotypes, but the authors don't provide any explanation as to why virulence is increased and why lipids are overproduced in these mutant strains. It seems to me that this could be the subject of further studies and that in the context of this manuscript it is either too much or not enough.

Response: While we acknowledge the reviewer's concerns, we have chosen to retain these results as they show that the changes in gene expression resulting from the loss of asymmetric methylation have an impact on the phenotype. We have revised the manuscript to emphasize that additional analyses are necessary to determine the underlying cause of the altered phenotypes (lines 507-508) and the corresponding figures depicted in Supplementary material.

Minor comments:

- Line 469: How many representatives of Zoopagomycota and Monoblepharomycota does this represent?

Response: The analysis includes two representatives of Zoopagomycota and one Monoblepharomycete. This information has also been indicated in the main text to enhance clarity (lines 99-100).

- Line 569: Does this mean that there is one MAC per gene? In this case it would mean that all genes are epigenetically marked to be expressed by default?

Response: The number of MAC detected for *P. blakesleeanus* NRRL 1555 is 9843 whereas the number of annotated genes in the version of the genome used is 16528. Also, some genes harbor more than one MAC as indicated in Figure 3D. A total of 6543 genes harbors at least one MAC, which makes up 40 % of genes. To enhance clarity in the text, we have added the percentage of genes with MACs in the text for both *Phycomyces* and *Mucor* (lines 180-182).

- Line 590: Why is this enzymatic activity considered common and enriched? Is this demonstrated by this study or previous work (citation required in this case)?

Response: This sentence refers to the GO analyses indicated in the same section and shown in Supplementary Figure 5. Pyrophosphate hydrolysis appears as an enriched function in the methylated genes from both *Mucor* and *Phycomyces*. To avoid misunderstandings, we have added a reference to the Figure in the text after this sentence (lines 196-197).

- Line 593: Is functional conservation of 6mA profiles an overstatement as only two species are compared, one of which shows so few 6mA?

Response: This refers to *M. lusitanicus* and *R. irregularis* and not *P. blakesleeanus*. A previous publication focused on *R. irregularis* (Chaturvedi et al 2021) (also low 6mA) analyzed the functional roles of 6mA in this fungus and found hydrolase and binding functions as enriched in methylated genes. Therefore, these are two different EDF representatives from different subphyla, and both display low 6mA levels, but the function associated with the methylated genes shows significant similarity. We have modified this section of the text to clarify this point (lines 197-199).

- Line 778: The SDs show that there is no significant difference in growth. Therefore, the phrase "...with a subtle increase during light growth, ..." does not seem appropriate.

Response: We agree with this comment, and we have eliminated the misleading sentence from the revised manuscript.

- Line 816-817: If the 6mA modification is not necessary for development, is it possible that it is important for fitness? Would it be possible to test this hypothesis?

Response:

This is an interesting point that warrants further investigation. Measuring fitness in filamentous fungi poses challenges, as determining survival and reproduction is complicated by the definition of the individual, as well as by the intricate and unusual life cycles and genetics involved (Pringle and Taylor, 2002. Trends Microbiol. 10:474-481. doi: 10.1016/s0966-842x(02)02447-2). Consequently, some authors choose to focus on a single aspect of the complex life cycle or a single measure of fitness, such as the number of asexual spores. The absence of differences in growth and asexual spore production between mutants in *met* genes and the wild-type strain may suggest no difference in fitness under laboratory conditions. However, the loss of *metB* results in susceptibility to hydroxyurea and reduced virulence, indicating a reduction in fitness under environmental conditions other than those found in the lab. We avoid going into this point in the discussion so as not to add to the complexity of the manuscript.

- Line 835: This observation suggests that the regulation of expression by 6mA is both direct and indirect. Therefore, before concluding that there is no association between asymmetric 6mA and active gene expression, it might be interesting to investigate whether the AAAACA motif, which should be present at the TSS of the down-regulated set of genes, would be absent in the up-regulated one.

Response: This comment is intriguing and prompted us to investigate the presence of AAAACA motifs in differentially expressed genes in *metB* mutants. We found that there is not a significant difference in the presence of the motif in upregulated or downregulated genes, as shown in the panel below. Specifically, 40.5% of the upregulated and 41.9% of the downregulated contain this motif. This data, along with the observed demethylation both in upregulated and downregulated genes (Figure 6F), support no association between asymmetric 6mA and active gene expression.

- Line 988-989: This argument seems weak because it would be impossible to genetically transform mammalian cells whose genomes have a complex repertoire of epigenetic modifications.

Response: We agree that the argument presented here would need additional characterizations. It was included based on previous suggestions made by Mondo et al (Nat Genet. 2017, 49:964-968. doi: 10.1038/ng.3859), who stated, “We suspect that the discovery of adenine methylation being strongly associated with gene expression may explain some of the historic difficulty in genetically modifying early-diverging fungi”. Given the marked differences in the epigenetic landscape between *Mucor* and *Phycomyces*, as well as their opposite tractability, we decided to follow up on this discussion. However, due to the lack of experimental evidence, we have removed this argument in the revised manuscript.

REVIEWER #2:

In this study, authors have investigated the distribution of 6mA among the genomes of early-diverged fungi and narrowed their investigation targets mainly for two species. They found that the species with high 6mA levels distributed in gene-rich regions are associated with symmetric deposition hosting positively correlated expressed genes. In contrast, species with low 6mA levels exhibit asymmetric 6mA modifications with no obvious regulation of gene expression association. This also coincides with the enzymes in charge of methylation of DNA. The study is very interesting to have a large variety of contents. However, I have some concerns about the methodology, design and the main results interpretation.

Response: We appreciate that the reviewer finds our study very interesting and provides insightful comments to improve the accuracy of our manuscript. Please note that we have condensed the text to meet the journal's requirements, while preserving the core information.

Comment #1. First, the main part of the study is actually about the comparison of two species of fungi, not about general EDF. It is nice that the authors’ design has focused on specific two species with deeper focusing on the differences between two species, but this cannot be generalized. In the general EDF analysis, authors already have shown that the 6mA patterns vary a lot among EDF without generic pattern and even there is high degree of differences among the different species of same phylum. However, at the same time, authors have picked two fungal species and compare the differences. Obviously, authors should be very careful to generalize the finding. For me the study is a mixture of two different contents, but feels like enumeration due to the results that cannot be generalized. The title and abstract are very misleading compared to the actual contents authors have presented. I would strongly recommend authors to change the

title and abstract to be specific to two species of fungi, instead of making too much generalization and emphasizing for the whole EDF.

Response: We agree with the reviewer's comment, and as a result, the title, abstract, and the rest of the text have been revised to be more specific.

Comment #2. Second, for the materials and methods, there are some points to be clarified and improved. Especially for the RNAseq data analysis and normalization. If I follow the order of sentences, authors used both FeatureCounts and DEseq2. Although FeatureCounts can just generate raw transcripts abundance, authors have specified that FPKM was calculated. Then in the next sentence authors used DEseq2 where FPKM should not be taken for the analysis. DEseq2's input must be the raw counts, as DEseq2 will normalize the transcript abundance by own way which is not FPKM. Normally, in transcriptomics, you don't use FPKM at all if you are analysing data by DEseq2. Therefore, in the case I concern if the whole transcriptome analysis data and related interpretations of this study are based on wrong normalization and consequent outcome. If authors did mistake, then the transcriptome analysis should be re-conducted before the revision.

Response: We appreciate the useful comment for bringing up that the description of the transcriptome analysis is confusing. As mentioned, DEseq2 used raw counts as it uses its own normalization. We utilized FeatureCounts to obtain raw counts and then ran DEseq2. The generation of FPKM and TPM information was independent of DEseq2 and was not considered in the analyses. However, the way it was described in the Method section of the original manuscript was confusing, as we initially mentioned FPKM generation from FeatureCounts, and subsequently introduced the DEseq2 analyses. Therefore, we have completely rewritten this part to be more precise and avoid confusion (lines 659-672).

Comment #3. Another part of M&M is about the statistical analysis. Authors used parametric ANOVA as a default, and then used post-hoc depending on the heteroscedasticity. By the way, the test authors have applied is Tukey's HSD, not Tucky. Anyway, if the normality is not tested and confirmed, it is incorrect to use the parametric ANOVA. There was no indication of normality in the part. If some of the data do not have normality, then authors should have applied non-parametric approaches to test the hypotheses. This part should be properly improved. Also, lastly, it seems there are some data showed heteroscedasticity as authors did Games-Howell post-hoc. However, for the T-test, authors simply used Student's T test. Weren't there any unequal variances of data? If so, authors should have used Welch's T test.

Response: We want to thank the reviewer for raising this concern, as the information provided in the initial manuscript was not accurately presented. The normality of the analyzed data was tested with the Shapiro-Wilk test (α 0.05), but this crucial detail was missing from the statistical description. We have now addressed this by adding the information to the Material and Methods section of the revised manuscript (lines 832-833).

Additionally, the reference to Tuckey's HSD/Games Howell was an error that originated from the statistical description of another ongoing manuscript. We apologize for this, and it has been rectified by indicating that Tukey's HSD post-hoc test was performed (line 835).

Furthermore, in response to the question regarding the t-test, we confirm that Welch's test was used instead of the student t-test. This was utilized to compare the expression levels of methylated and unmethylated genes. Sample sizes were highly different in most cases. We have corrected this in the Material and Methods section of the revised manuscript (lines 837-840),

the legends of Fig. 4 (lines 1325-1326) and Supplementary Fig. 12C have also been updated accordingly to reflect the accurate information.

Comment #4. Third, for the 6mA occurrence and gene expression tests and the interpretation, it would be better to categorize the different occurring points of 6mA, before challenge the corresponding gene expression. It has been known that not only the symmetric or asymmetric 6mA affects the gene expression, but also the methylated sites (upstream, promoter, gene body, downstream) can differently affect the gene expression. Have more detailed expression information with each case will make authors interpretation more convincing for the readers. At the same time, since the 5mC occurrence was also investigated, authors can also provide more holistic overview of the methylation and gene expression relationships.

Response: As mentioned in this comment, the specific distribution of 6mA is important for its role in the regulation of gene expression. In our case, the clusters of 6mA (found in *Phycomyces*) show a strong association with active gene expression. Since these clusters usually appear around (mostly downstream) the TSS, we analyzed the association between 6mA and gene expression, considering both 6mA sites and clusters, using a window of -100 + 400bp from the TSS, as indicated in the legend to Figure 3. Downstream and/or intergenic 6mA is scarce (Fig. 2) and not clustered, so we focused on those regions in which 6mA is particularly enriched and follows a specific distribution pattern.

In the case of 5mC, as this modification is mostly found on repeats, we have kept the analyses separated in the revised manuscript. However, we have explored 5mC in both the top more highly and lower expressed genes, and since we have shown the relevance of 6mA presence in gene expression regulation, these two types of DNA methylation should have been better connected. To address this, we have included explanatory and connecting sentences in the section that described 5mC and gene expression in the “Genomic implications of DNA methylation in Mucorales” results section, highlighting the imbricated and contrasting role of both modifications in gene expression.

Comment #5. Forth, although authors have focused on the finding the homologs of methyl transferases associated with 6mA, the observed functions of homologs sometime are different than known homologs. Indeed, some DNA methyl transferases can work for both 5mC and 6mA. Therefore, it is difficult to really affirm that the observed difference in gene expression and phenotypic changes are solely from the changes in 6mA. Authors should clarify there was no change 5mC in each case.

Response: Indeed, this comment is appropriate. We conducted 5mC profiling for all mutants, but to avoid overwhelming the readers of the manuscript, we did not describe this in detail in the text. This information was included in Supplementary Table 2, but we didn't mention it in the manuscript. Since 5mC in *Mucor* is scarce and not CG enriched, we decided not to focus too much on it in this paper until we can conduct a specific investigation on its possible roles. Nevertheless, we thoroughly examined the methylation motifs and distribution in the genome, but we did not detect any relevant differences between the wild-type strain and the mutants (see panel below, MU636 is the wild-type strain, and it was compared to mutants' samples in nitrogen-depleted medium (-N) or light (L). Please, note that light enhances abundance in 5mC (Fig. 4A)). We aim to avoid providing extra information that may not be informative, so we do not plan to include this panel in the final article. However, as we agree that this consideration should be addressed in the text, we have included references to Supplementary Table 2

summarizing this information and clarified that no relevant changes in 5mC were detected (lines 371-372).

Comment #6. Fifth, about the HGT, authors stated that the MetB was only conserved in certain lineages listed in the manuscript, but with exceptions. In actual data, there are other basal lineages having the enzyme. Clearly, the enzyme occurred from the basal lineages of Fungal kingdom, or even more ancient ancestors in the tree of life. At the same time, the information based on ProFam database is mainly biased by bacteria due to functional characterization. In this case, can author really say that the enzyme is originated “from” bacteria “to” fungi? What makes authors to only consider one direction, or the enzymes have a same origin of HGT? For me, it is very difficult to make HGT claim without proper analysis. Proper phylogenetic analysis should be done for the claim. Does the gene have different codon usage? Or Does synteny of the genome have any indication of HGT? I think the claim should be carefully addressed with clear data.

Response: We agree with the reviewer that the question of HGT is generally complex. However, our phylogenetic tree (Supplementary File 5 displays the phylogenetic tree showing all sequence names, support values and NCBI accessions) indicates placement of two MetB fungal clades within a bacterial tree. Both fungal clades are well supported. The diversity of sequences in bacteria is greater than in each of the fungal clade, suggesting that diversification occurred in bacteria and two single HGT events gave origin to fungal methyltransferases (clades 1 and 2). None of fungal clades are recent enough to exhibit signs of HGT in sequence terms, and they resemble fungal genes, including the presence of introns. Phylogenetic incongruence and unexpected levels of sequence similarity are the current clues supporting the HGT scenario. These are considered as the strongest line of evidence (Fitzpatrick 2012. FEMS Microbiol Lett. 329:1-8. doi: 10.1111/j.1574-6968.2011.02465.x). The alternative scenario would entail a massive loss of the methylase in all non-fungal eukaryotes twice in evolution.

In our case, as well as in other old HGT the GC content or codon usage variation between the xenolog and the genomic background disappear because of amelioration over time (Fitzpatrick 2012. FEMS Microbiol Lett. 329:1-8. doi: 10.1111/j.1574-6968.2011.02465.x). Ancient HGT of methylases is documented in literature (Iyer et al., 2016. *Bioessays* 38: 27-40. doi: 10.1002/bies.201500104)

To reduce the assertiveness of the statement about HGT from bacteria to fungi, the text in the manuscript has been revised (lines 508-516).

Comment #7. For the discussion part, I think the discussion can be carefully re-addressed after the additional data analyses and checking. I just have few minor comments in discussion.

L1081-1082 – It seems EDF-bacteria HGT was also reported in other fungi. PMID: 29329439

Response: We have thoroughly reviewed the discussion and incorporated the suggested reference. Thank you for bringing this work to our attention. Initially, we only considered HGT involving metabolism, but this reference expands our observations and enriches the discussion.

REVIEWER #3:

The manuscript entitled “Symmetric and asymmetric DNA 6-methyladenine regulates different biological responses in early-diverging fungi” by Lax et al. impressively unravels the presence and function of an underrepresented and understudied epigenetic modification, DNA adenine methylation (6mA), in an equally understudied group of fungi, the early-diverging fungi (EDF). EDF were identified to have comparatively higher levels of 6mA than Dikarya or higher eukaryotes, making them promising model organisms to study this modification.

The authors have chosen two different EDF species for their comparative epigenetic analysis under different physiological conditions – a very powerful experimental setup – as these are characterized by different DNA methylation profiles: While *Phycomyces* (*P. blakesleeanus* NRRL1555) has 1.13% of adenines methylated, *Mucor* (*M. lusitanicus* CBS 277.49) has only 0.26% of adenines methylated, as quantified by Pacbio SMRT sequencing. *Phycomyces* adenines are symmetrically methylated, while *Mucor* adenines are asymmetrically methylated. Additionally, *Phycomyces* has high levels of 5mC, while *Mucor* again has low levels of this DNA modification. In *Phycomyces*, there is a good correlation between 5mC and gene repression, as well as 6mA and gene activation (likely due to the symmetric modification), while this is not the case in *Mucor*.

The authors then go on to delete 6mA DNA methyltransferase candidate genes, and thereby identify and characterize the responsible enzymes for both the symmetric and asymmetric modification, suggesting heterologous gene transfer for the rare asymmetric 6mA methyltransferase MetB.

Overall, this manuscript is well thought out, well researched and written, and was a pleasure to read. The following comments are intended to make some of the conclusions possibly more clear and stronger:

Response: We appreciate that the reviewer finds our study interesting and well conducted and provides relevant comments to improve our manuscript. Please note that we have condensed the text to meet the journal's requirements, while preserving the core information.

Comment #1. Line 545: Does the expression of DNA methyltransferase genes match the observed (different) levels of 5mC and 6mA in the two analyzed EDF species?

Response: This is a very interesting question that we also pondered during our investigation. Given that the components are conserved in both species, we speculated that we could detect differences at expression level. However, we found that in both species, the expression of the genes involved in both types of methylation occurs within a similar range, with no significant differences or instances of silencing in one of the species (see panels below). This suggests that there must be additional complexities at play. Furthermore, all these proteins are expressed at relatively low levels compared to the total population of genes, which is characteristic of

regulatory proteins. Now that these components have been identified in this investigation, our aim is to further characterize how they function in the fungal kingdom to unravel their complexity and gain valuable insights into why we observe differences among species that cannot be explained by conservation or expression level.

	MetB Mucor 1597305	MetB Phycomyces 182778	P1 Mucor 1569123	P1 Phycomyces 157782	Mta1 Mucor 1550399	Mta1 Phycomyces 172632	Mta9 Mucor 1371135	Mta9 Phycomyces 138029
Average TMM Mycellum Light	21,64666667	26,6	38,35333333	21,20666667	31,99	27,86	61,81	36,43666667
Average TMM Mycellum Dark	21,65333333	19,73666667	40,47666667	20,53	30,67	25,24666667	58,84333333	37,49666667

Comment #2. Line 595: Did you notice this in *Mucor*, and could you check whether in your deletion mutants the expression level of the respective 5mC-specific genes is affected?

Response: We also investigated the potential for this cross-regulation in *Mucor*, but unfortunately, we did not find this enrichment. However, the likelihood of coregulation diminished due to the scarcity of 5mC in *Mucor*, unlike in *Phycomyces*. Furthermore, we conducted bisulfite-seq on *Mucor* mutants and did not detect any significant changes in the levels and distribution of 5mC (see response to Comment #5 from reviewer #2).

Comment #3. Line 810: Going in line with the previous comment - symmetric 6mA is increased in the knockout of *metB*, which catalyzes the asymmetric modification. Would this regulation be performed on a transcriptional level?

Response: The reviewer's comment highlights confusion regarding the information provided. Therefore, the text has been revised to clearly convey that the proportion of symmetric 6mA increases in the *metB* knockout due to the reduction of asymmetric 6mA, rather than an absolute increase in symmetric 6mA (lines 368-369).

Comment #4. Line 822: How do you explain the reduced virulence of the knockout of *metC*, and even increased virulence of the knockout of *metA*, while they are not contributing to 6mA levels? What do you suggest for their function?

Response: These genes encode putative DNA 6mA methyltransferases, but their domains have also been identified in proteins known to methylate RNA, suggesting potential involvement in RNA methylation. The domain present in *MetA* is also found in methyltransferase C (e.g., Swiss:P44453) and other methyltransferases (e.g., Swiss:Q53742). Similarly, the domain present in *MetC* is found in the S-adenosylmethionine-binding subunit of human mRNA:m6A methyltransferase (MTase), an enzyme responsible for methylation of adenines in pre-mRNAs. Determining the function of these proteins in EDF requires extensive work beyond the scope of the current manuscript, which will be undertaken in the future.

Comment #5. Line 866: Unfortunately, this contradicts your narrative that symmetric 6mA is correlated with active gene expression. Do you have any hypothesis why *Mucor* might not fit your hypothesis?

Response: This sentence is clearly misleading, and it has been rephrased in the revised manuscript (lines 408-410). In *Mucor*, there are low levels of symmetric 6mA, with the majority not concentrated in MACs, which are linked to active gene expression. The fact that *Mucor* almost completely lacks these high-density 6mA clusters downstream of the TSS could explain that other regulatory mechanisms have arisen to supply the symmetric 6mA function in EDF instead of relying on the very few symmetric sites found on its genome.

Minor comments:

- Line 521: Please make a connection to symmetric/asymmetric modifications, which is only mentioned in the next paragraph.

Response: Addressed. We have linked both concepts in that section by including two sentences (lines 139-140 and 150).

- Line 531: When analyzing

Response: Changed.

- Line 543: Is this relevant, is this complex conserved in EDF?

Response: We did not find this complex conserved either in *Mucor* or *Phycomyces*. Given our results in which *metB* deletion leads to the loss of asymmetric sites, we hypothesize that MetB protein is responsible for this modification. In contrast, asymmetric deposition of 6mA in more complex eukaryotes is controlled by a wider range of methyltransferases. This is evident with MetA, the fungal protein with high similarity to the human protein involved in asymmetric 6mA deposition, which is not responsible for DNA 6mA methylation in EDF. Therefore, a functional diversification and expansion of 6mA methyltransferases is the most probable explanation.

- Typo in legend of figure 4: 5 6mC

Response: Corrected. Thank you.

REVIEWERS' COMMENTS

Reviewer #1 (Remarks to the Author):

I would like to thank the authors for their work. I am satisfied with the answers to my questions and therefore recommend publication of the revised manuscript.

Reviewer #2 (Remarks to the Author):

Authors have revised the manuscript titled "Symmetric and asymmetric DNA N6-adenine methylation regulates different biological responses in Mucorales". In revised version, the main title was changed to appropriately reflect the content of the manuscript. I appreciate authors efforts.

Intra-Genus level comparison between two mucoromycota species to represent whole EDF? Although the revised version amended many statements to avoid generalization of main findings from two fungal species methylomes to be general EDF-related finding, still there are multiple descriptions throughout the manuscript which can confuse readers.

Also, authors put efforts to maximize the value of investigating two fungal species for representing EDF. In line 169-174, authors have added statements that the one species of Mucoromycota has similar 5mC feature as one Zoopagomycotan fungal species, emphasizing that this might expand the species-focused finding to the general EDF. However, as mentioned in the Line 101-106, these are only 2 fungal species from 2 phyla among 10 EDF phyla presented. Moreover, the highlighted EDF phyla to be epigenetically differed/varied from Mucoromycota were actually not Zoopagomycota. At the same time, as authors presented, there is a substantial epigenetic variation among EDF. It is very concerning as using two fungal species methylome investigation to reflect other cases of EDF will be insufficient. Although authors have revised to not generalize their findings, still some cases, it should be clarified that these are mainly about two fungal species.

Reviewer #3 (Remarks to the Author):

All of my comments have been sufficiently addressed.

REVIEWER COMMENTS

REVIEWER #2:

Authors have revised the manuscript titled “Symmetric and asymmetric DNA N6-adenine methylation regulates different biological responses in Mucorales”. In revised version, the main title was changed to appropriately reflect the content of the manuscript. I appreciate authors efforts.

Intra-Genus level comparison between two mucoromycota species to represent whole EDF? Although the revised version amended many statements to avoid generalization of main findings from two fungal species methylomes to be general EDF-related finding, still there are multiple descriptions throughout the manuscript which can confuse readers.

Also, authors put efforts to maximize the value of investigating two fungal species for representing EDF. In line 169-174, authors have added statements that the one species of Mucoromycota has similar 5mC feature as one Zoopagomycotan fungal species, emphasizing that this might expand the species-focused finding to the general EDF. However, as mentioned in the Line 101-106, these are only 2 fungal species from 2 phyla among 10 EDF phyla presented. Moreover, the highlighted EDF phyla to be epigenetically differed/varied from Mucoromycota were actually not Zoopagomycota. At the same time, as authors presented, there is a substantial epigenetic variation among EDF. It is very concerning as using two fungal species methylome investigation to reflect other cases of EDF will be insufficient. Although authors have revised to not generalize their findings, still some cases, it should be clarified that these are mainly about two fungal species.

Response: We acknowledge the reviewer's concerns about generalization and have thoroughly revised the manuscript to rewrite all text to avoid drawing unsupported conclusions from the results. These changes concern lines 94, 171-172, 175, 349, 395, 425-426, 513, and 529 of the revised manuscript.